# Ambient Diffusion Omni:
# Training Good Models with Bad Data

## Abstract

We show how to use low-quality, synthetic, and out-of-distribution images to improve the quality of a diffusion model. Typically, diffusion models are trained on curated datasets that emerge from highly filtered data pools from the Web and other sources. We show that there is immense value in the lower-quality images that are often discarded. We present Ambient Diffusion Omni, a simple, principled framework to train diffusion models that can extract signal from all available images during training. Our framework exploits two properties of natural images – spectral power law decay and locality. We first validate our framework by successfully training diffusion models with images synthetically corrupted by Gaussian blur, JPEG compression, and motion blur. We then use our framework to achieve state-of-the-art ImageNet FID and we show significant improvements in both image quality and diversity for text-to-image generative modeling. The core insight is that noise dampens the initial skew between the desired high-quality distribution and the mixed distribution we actually observe. We provide rigorous theoretical justification for our approach by analyzing the trade-off between learning from biased data versus limited unbiased data across diffusion times.

## 1 Introduction

Large-scale, high-quality training datasets have been a primary driver of recent progress in generative modeling. These datasets are typically assembled by filtering massive collections of images sourced from the web or proprietary databases [25, 43, 53, 58, 59]. The filtering process is crucial to the quality of the resulting models [13, 27, 25, 32, 27]. However, filtering strategies are often heuristic and inefficient, discarding large amounts of data [51, 43, 25, 13]. We demonstrate that the data typically rejected as low-quality holds significant, underutilized value.

Extracting meaningful information from degraded data requires algorithms that explicitly model the degradation process. In generative modeling, there is growing interest in approaches that learn to generate directly from degraded inputs [18, 17, 14, 15, 7, 47, 39, 52, 5, 1, 2, 55, 71, 46, 64, 45, 11, 48]. A key limitation of existing methods is their reliance on knowing the exact form of the degradation. In real-world scenarios, image degradations—such as motion blur, sensor artifacts, poor lighting, and low resolution—are often complex and lack a well-defined analytical description, making this assumption unrealistic. Even within the same dataset, from ImageNet to internet scale text-to-image datasets, there are samples of varying qualities [28], as shown in Figures 3, 24, 27, 25. Given access to this mixed-bag of datapoints, we would like to sample from a tilted continuous measure of high-quality images, without sacrificing the diversity present in the training points.

The training objective of diffusion models naturally decomposes sampling from a target distribution into a sequence of supervised learning tasks [30, 61, 62, 16, 19, 9, 10]. Due to the power-law structure

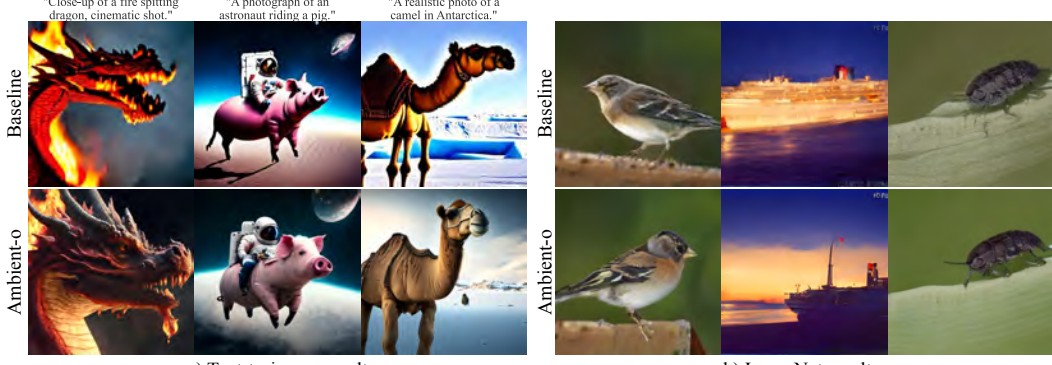

a) Text-to-image results        b) ImageNet results

Figure 1: Effect of using Ambient-o for (a) training a text-to-image model (Micro-Diffusion [54]) and (b) a class-conditional model for ImageNet (EDM-2 [35]). All generations are initialized with the same noise. The baseline models are trained using all the data equally. Ambient-o changes the way the data is used during the diffusion process based on its quality. This leads to significant visual improvements without sacrificing diversity, as would happen with a filtering approach (see Fig. 29).

of natural image spectra [65], high diffusion times focus on generating globally coherent, semantically meaningful content [22], while low diffusion times emphasize learning high-frequency details.

Our first key theoretical insight is that low-quality samples can still be valuable for training in the high-noise regime. As noise increases, the diffusion process contracts distributional differences (see Theorem 4.2), reducing the mismatch between the high-quality target distribution and the available mixed-quality data. At the same time, incorporating low-quality data increases the sample size, reducing the variance of the learned estimator. Our analysis formalizes this bias–variance trade-off and motivates a principled algorithm for training denoisers at high diffusion times using noisy, heterogeneous data.

For low diffusion times, our algorithm leverages a second key property of natural images: locality. We show a direct relationship between diffusion time and the optimal receptive field size for denoising. Specifically, small image crops suffice at lower noise levels. This allows us to borrow high-frequency details from out-of-distribution or synthetic images, as long as the marginal distributions of the crops match those of the target data.

We introduce Ambient Diffusion Omni (Ambient-o), a simple and principled framework for training diffusion models using arbitrarily corrupted and out-of-distribution data. Rather than filtering samples based on binary 'good' or 'bad' labels, Ambient-o retains all data and modulates the training process according to each sample's utility. This enables the model to generate diverse outputs without compromising image quality. Empirically, Ambient-o advances the state of the art in unconditional generation on ImageNet and enhances diversity in text-conditional generation without sacrificing fidelity. Theoretically, it achieves improved bounds for distribution learning by optimally balancing the bias–variance trade-off: low-quality samples introduce bias, but their inclusion reduces variance through increased sample size.

## 2 Background and Related Work

**Diffusion Modeling.** Diffusion models transform the problem of sampling from $p_0$ into the problem of learning *denoisers* for smoothed versions of $p_0$ defined as $p_t = p_0 \circledast \mathcal{N}(0, \sigma^2(t)\mathrm{I})$. We typically denote with $X_0 \sim p_0$ the R.V. distributed according to the distribution of interest and $X_t = X_0 + \sigma(t)Z$, the R.V. distributed according to $p_t$. The target is to estimate the set of optimal $l_2$ denoisers, i.e., the set of the conditional expectations: $\{\mathbb{E}[X_0|X_t = \cdot]\}_{t=1}^{T}$. Typically, this can be achieved through supervised learning by minimizing the following loss (or a re-parametrization of it):

$$J(\theta) = \mathbb{E}_{t \in \mathcal{U}[0,T]} \mathbb{E}_{x_0, x_t|t} \left[ ||h_\theta(x_t, t) - x_0||^2 \right], \tag{2.1}$$

that is optimized over a function family $\mathcal{H} = \{h_\theta : \theta \in \Theta\}$ parametrized by network parameters $\theta$. For sufficiently expressive families, the minimizer is indeed: $h_{\theta*}(x, t) = \mathbb{E}[X_0|X_t = x]$.

**Learning from noisy data.** The diffusion modeling framework described above assumes access to samples from the distribution of interest $p_0$. An interesting variation of this problem is to learn to sample from $p_0$ given access to samples from a tilted measure $\tilde{p}_0$ and a known degradation model. In Ambient Diffusion [18], the goal is to sample from $p_0$ given pairs $(Ax_0, A)$ for a matrix $A : \mathbb{R}^{m \times n}, m < n$, that is distributed according to a known density $p(A)$. The techniques in this work were later generalized to accommodate additive Gaussian Noise [15, 17, 1] in the measurements. More recently there have been efforts to further broaden the family of degradation models considered through Expectation-Maximization approaches that involve multiple training runs [52, 5].

Recent work from [17] has shown that, at least for the Gaussian corruption model, leveraging the low-quality data can tremendously increase the performance of the trained generative models. In particular, the authors consider the setting where we have access to a few samples from $p_0$, let's denote them $\mathcal{D}_0 \{x_0^{(i)}\}_{i=1}^{N_1}$ and many samples from $p_{t_n}$, let's denote them $\mathcal{D}_{t_n} \{x_{t_n}^{(i)}\}_{i=1}^{N_2}$, where $p_{tn} = p_0 \circledast \mathcal{N}(0, \sigma^2(t_n)\mathrm{I})$ is a smoothed version of $p_0$ at a known noise level $t_n$. The clean samples are used to learn denoisers for all noise levels $t \in [0, T]$ while the noisy samples are used to learn denoisers only for $t \geq t_n$, using the training objective:

$$J_{\mathrm{ambient}}(\theta) = \mathbb{E}_{t \in \mathcal{U}(t_n, T]} \sum_{i=1}^{N_2} \mathbb{E}_{x_t | x_{t_n}^{(i)}} \left[ \left|\left| \alpha(t) h_\theta(x_t, t) + (1 - \alpha(t))x_t - x_{t_n}^{(i)} \right|\right|^2 \right], \qquad (2.2)$$

with $\alpha(t) = \frac{\sigma^2(t) - \sigma^2(t_n)}{\sigma^2(t)}$. Note that the objective of equation 2.2 only requires samples from $p_{t_n}$ (instead of $p_0$) and can be used to train for all times $t \geq t_n$. This algorithm uses $N_1 + N_2$ datapoints to learn denoisers for $t > t_n$ and only $N_1$ datapoints to learn denoisers for $t \leq t_n$. The authors show that even for $N_1 << N_2$, the model performs similarly to the setting of training with $(N_1 + N_2)$ clean datapoints. The main limitation of this method and its related works is that the degradation process needs to be known. However, in many applications, we have data from heterogeneous sources and various qualities, but there is no analytic form or any prior on the corruption model.

**Data filtering.** One of the most crude, but widely used, approaches for dealing with heterogeneous data sources is to remove the low-quality data and train only the high-quality subset [43, 25, 23]. While this yields better results than naively training on the entire distribution, it leads to a decrease in diversity and relies on heuristics for optimizing the filtering. An alternative strategy is to train on the entire distribution and then fine-tune on high-quality data [13, 54]. This approach better trades the quality-diversity trade-off but still incurs a loss of diversity and is hard to calibrate.

**Training with synthetic data.** A lot of recent works have shown that synthetic data can improve the generative capabilities of diffusion models when mixed properly with real data from the distribution of interest [24, 3, 4]. In this work, we show that it helps significantly to view synthetic data as corrupted versions of the samples from the real distribution and incorporate this perspective into the training objective.

# 3 Method

We propose a new framework that extends beyond [17] to enable training generative models directly from arbitrarily corrupted and out-of-distribution data, without requiring prior knowledge of the degradation process. We begin by formalizing the setting of interest.

**Problem Setting.** We are given a dataset $\mathcal{D} = \{w_0^{(i)}\}_{i=1}^N$ consisting of $N$ datapoints. Each point in $\mathcal{D}$ is drawn from a mixture distribution $\tilde{p}_0$, which mixes $p_0$ (the distribution of interest) and an alternative distribution $q_0$ that may contain various forms of degradation or out-of-distribution content. We assume access to two labeled subsets, $S_G, S_B$, where points in $S_G$ are known to come from the clean distribution $p_0$, and points in $S_B$ from the corrupted distribution $q_0$. While this assumption simplifies the initial exposition, we relax it in Section F.1. We focus on the practically relevant regime where $|S_G| \ll |\mathcal{D}|$—i.e., access to high-quality data is severely limited. The objective is to learn a generative model that (approximately) samples from the clean distribution $p_0$, leveraging both clean and corrupted samples in its training.

We now describe how degraded and out-of-distribution samples can be effectively leveraged during training in both the high-noise and low-noise regimes of the diffusion process.

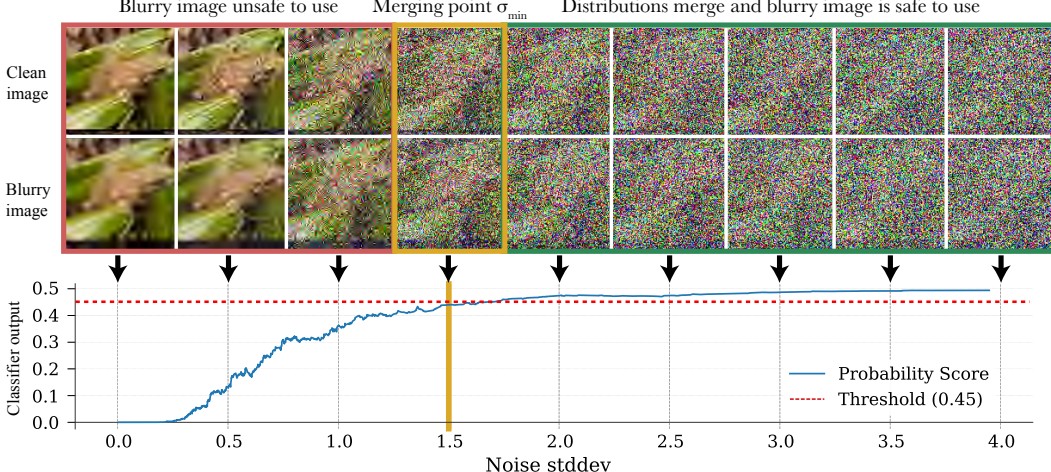

Figure 2: **A time-dependent classifier trained to distinguish noisy clean and blurry images** (blur kernel standard deviation $\sigma_B = 0.6$). At low noise the classifier is able to perfectly identify the blurry images, and outputs a probability close to 0. As the noise increases and the information in the image is destroyed, the clean and blurry distributions converge and the classifier outputs a prediction close to 0.5. The red line plots the threshold (selected at $\tau = 0.45$), which is crossed at $\sigma_t = 1.64$.

### 3.1 Learning in the high-noise regime (leveraging low-quality data)

**Addition of gaussian noise contracts distribution distances.** The first key idea of our method is that, at high diffusion times $t$, the noised target distribution $p_t$ and the noised corrupted distribution $\tilde{p}_t$ become increasingly similar (Theorem 4.2), effectively attenuating the discrepancy introduced by corruption. This effect is illustrated in Figure 2 (top), where we compare a clean image and its degraded counterpart (in this case, corrupted by Gaussian blur). As the diffusion time $t$ increases, the noised versions of both samples become visually indistinguishable. Consequently, samples from $\tilde{p}_0$ can be leveraged to learn (the score of) $p_t$, for $t > t_n^{\min}$. We formalize this intuition in Section 4, and we also quantify that for large $t$ there are statistical efficiency benefits for using a large sample from $\tilde{p}_0$ versus a small sample from $p_0$.

**Heuristic selection of the noise level.** From the discussion so far, it follows that to use samples from $\tilde{p}_0$, we need to assign them to a noise level $t_n^{\min}$. One can select this noise level empirically, i.e. we can ablate this parameter by training different models and selecting the one that maximizes the generative performance. However, this approach requires multiple trainings, which can be costly. Instead, we can find the desired noise level in a principled way as detailed below.

**Training a classifier under additive Gaussian noise.** To identify the appropriate noise level, we train a time-conditional classifier to distinguish between the noised distributions $p_t$ and $q_t$ across various diffusion times $t$. We use a single neural network $c_\theta^{\mathrm{noise}}(x_t, t)$ that is conditioned on the diffusion time $t$, following the approach of time-aware classifiers used in classifier guidance [21]. The classifier is trained using labeled samples from $S_G$ (clean) and $S_B$ (corrupted) via the following objective:

$$J_{\mathrm{noise}}(\theta) = \sum_{x_0 \in S_G} \mathbb{E}_{x_t|x_0} \left[ -\log c_\theta^{\mathrm{noise}}(x_t, t) \right] + \sum_{y_0 \in S_B} \mathbb{E}_{y_t|y_0} \left[ -\log(1 - c_\theta^{\mathrm{noise}}(y_t, t)) \right] \quad (3.1)$$

**Annotation.** Once the classifier is trained, we use it to determine the minimal level of noise that must be added to the low-quality distribution $q_0$ so that it closely approximates a smoothed version of the high-quality distribution $p_0$. Formally, we compute:

$$t_n^{\min} = \inf \left\{ t \in [0, T] : \frac{1}{|S_B|} \sum_{y_0 \in S_B} \mathbb{E}_{y_t|y_0} \left[ c_\theta^{\mathrm{noise}}(y_t, t) \right] > \tau \right\}, \quad (3.2)$$

for $\tau = 0.5 - \epsilon$ and for some $\epsilon > 0$. Subsequently, we form the annotated dataset $\mathcal{D}_{\overline{\mathrm{annot}}} = \{(w_0^{(i)} + \sigma_{t_n^{\min}} Z^{(i)}, t_n^{\min})\}_{i=1}^N \cup \{(x_0, 0) | x_0 \in S_G\}$, where the random variables $Z^{(i)}$ are i.i.d. standard

normals. In particular, our annotated dataset indicates that we should only use the samples from $\mathcal{D}$ for diffusion times $t \geq t_n^{\min}$, for which the distributions have approximately merged and hence it is safe to use them. In fact, the optimal classifier assigns time $t_n$ that corresponds to the first time for which $\mathrm{d}_{\mathrm{TV}}(p_t, q_t) \leq \epsilon$.

**From arbitrary corruption to additive Gaussian noise.** The afore-described approach reduces our problem of learning from data with arbitrary corruption to the setting of learning from data corrupted with additive Gaussian noise. The price we pay for this reduction is the information loss due to the extra noise we add to the samples during the annotation stage. We can now extend the objective function (2.2) to train our diffusion model. Suppose our annotated dataset is comprised of samples $\{(x_{t_i^{\min}}^{(i)}, t_i^{\min})\}$. Then our objective becomes:

$$J_{\text{ambient}-\text{o}}(\theta) = \mathbb{E}_{t \in \mathcal{U}[0,T]} \sum_{i: t_i^{\min} < t} \mathbb{E}_{x_t | x_{t_i^{\min}}^{(i)}} \left[ \left\| \left| \alpha(t, t_i^{\min}) h_\theta(x_t, t) + (1 - \alpha(t, t_i^{\min})) x_t - x_{t_i^{\min}}^{(i)} \right| \right\|^2 \right],$$

where $\alpha(t, t_i^{\min}) = \frac{\sigma^2(t) - \sigma^2(t_i^{\min})}{\sigma^2(t)}$.

Moreover, the method is particularly well-suited to certain types of corruptions but is less effective for others. Because the addition of Gaussian noise suppresses high-frequency components—due to the spectral power law of natural images—our approach is most effective for corruptions that primarily degrade high frequencies (e.g., blur). In contrast, degradations that affect low-frequency content—such as color shifts, contrast reduction, or fog-like occlusions—are more challenging. This limitation is illustrated in Figure 9: masked images, for example, require significantly more noise to become usable compared to high-frequency corruptions like blur. In the extreme, the method reduces to a filtering approach, as infinite noise nullifies all information in the corrupted samples.

## 3.2 Learning in the low-noise regime (synthetic and out-of-distribution data)

So far, our algorithm implicitly results in varying amounts of training data across diffusion noise levels. At high noise, the model can leverage abundant low-quality data, whereas at low noise levels, it must rely solely on the limited set of high-quality samples. We now extend the algorithm to enable the use of synthetic and out-of-distribution data for learning denoisers at low-noise diffusion times.

To achieve this, we leverage another fundamental property of natural images: *locality*. At low diffusion times, the denoising task can be solved using only a small local region of the image, without requiring full spatial context. We validate this hypothesis experimentally in the Experiments Section (Figures 11, 12, 13, 14), where we show that there is a mapping between diffusion time $t$ and the crop size needed to perform the denoising optimally at this diffusion time. Intuitively, the higher the noise, the more context is required to accurately reconstruct the image. Conversely, for lower noise, the local information within a small neighborhood suffices to achieve effective denoising. We use $\mathrm{crop}(t)$ to denote the minimal crop size needed to perform optimal denoising at time $t$. If there are two distributions $p_0$ and $\tilde{p}_0$ that agree on their marginals (i.e. crops), they can be used interchangeably for low-diffusion times. Note that the distributions don't have to agree globally, they only have to agree on a local (patch) level. Formally, let $A(t)$ be a random patch selector of size $\mathrm{crop}(t)$. Let also $p_0, \tilde{p}_0$ two distributions that satisfy:

$$A(t)\#p_0 = A(t)\#\tilde{p}_0, \tag{3.3}$$

where $A(t)\#p_0$ denotes the pushforward measure[1] of $p_0$ under $A(t)$. Then, the cropped portions of the tilted distributions provide equivalent information to the crops of the original distribution for denoising.

**Training a crops classifier.** Note that the condition of Equation (3.3) can be trivially satisfied if $A(t)$ masks all the pixels or even if $A(t)$ just selects a single pixel. We are interested in finding what is the maximum crop size for which this condition is approximately true. Once again, we can use a classifier to solve this task. The input to the classifier, $c_\theta^{\text{crops}}$, is a crop of an image that either arises from $p_0$ or $\tilde{p}_0$, and the classifier needs to classify between these two cases.

---

[1] Given measure spaces $(X_1, \Sigma_1)$ and $(X_2, \Sigma_2)$, a measurable function $f : X_1 \to X_2$, and a probability measure $p : \Sigma_1 \to [0, \infty)$, the pushforward measure $f\#p$ is defined as $(f\#p)(B) := p(f^{-1}(B)) \ \forall B \in \Sigma_2$.

**Annotation and training using the trained classifier.** Once the classifier is trained, we are now interested in finding the biggest crop size for which the distributions $p_0, \tilde{p}_0$ cannot be confidently distinguished. Formally,

$$t_n^{\max} = \sup \left\{ t \in [0, T] : \frac{1}{|S_B|} \sum_{y_0 \in S_B} [c_\theta^{\text{crops}}(A(t)(y_t))] > \tau \right\}, \tag{3.4}$$

for $\tau = 0.5 - \epsilon$ and for some small $\epsilon > 0$[2]. For times $t \leq t_n^{\max}$, the out-of-distribution images from $\tilde{p}_0$ can be used with the regular diffusion objective as images from $p_0$, as for these times the denoiser only looks at crops and at the crop level the distributions have converged.

**The donut paradox.** Each sample can be used for $t \geq t_i^{\min}$ and for $t \leq t_i^{\max}$, but not for $t \in (t_i^{\max}, t_i^{\min})$. We call this the *donut paradox* as there is a hole in the middle of the diffusion trajectory for which we have fewer available data. These times do not have enough noise for the distributions to merge globally, but also the required receptive field for denoising is big enough so that there are differences on a crop level. We show an example of this effect in Figure 10.

# 4 Theory

We study the 1-d case, but all our claims easily extend to any dimension. We compare two algorithms:

**Algorithm 1.** Algorithm 1 trains a diffusion model using access to $n_1$ samples from a target density $p_0$, assumed to be supported in $[0, 1]$ and be $\lambda_1$-Lipschitz.

**Algorithm 2.** Algorithm 2 trains a diffusion model using access to $n_1 + n_2$ samples from a density $\tilde{p}_0$ that is a mixture of the a target density $p_0$ and another density $q_0$, assumed to be supported in $[0, 1]$ and be $\lambda_2$-Lipschitz: $\tilde{p}_0 = \frac{n_1}{n_1+n_2} p_0 + \frac{n_2}{n_1+n_2} q_0$.

We want to compare how well these algorithms estimate the distribution $p_t := p_0 \circledast \mathcal{N}(0, \sigma_t^2)$. We use $\hat{p}_t^{(1)}, \hat{p}_t^{(2)}$ to denote the estimates obtained for $p_t$ by Algorithms 1 and 2 respectively.

**Diffusion modeling is Gaussian kernel density estimation.** We start by making a connection between the optimal solution to the diffusion modeling objective and kernel density estimation. Given a finite dataset $\{W^{(i)}\}_{i=1}^n$, the optimal solution to the diffusion modeling objective should match the empirical density at time $t$, which is:

$$\hat{p}_t(x) = \frac{1}{n\sigma_t} \sum_{i=1}^n \phi\left(\frac{W^{(i)} - x}{\sigma_t}\right), \tag{4.1}$$

where $\phi(u) = \frac{1}{\sqrt{2\pi}} e^{-u^2/2}$ is the Gaussian kernel. We observe that equation 4.1 is identical to a Gaussian kernel density estimate, given samples $\{W^{(i)}\}_{i=1}^n$[3].

We establish the following result for Gaussian kernel density estimation.

**Theorem 4.1** (Gaussian Kernel Density Estimation)**.** *Let $\{W^{(i)}\}_{i=1}^n$ be a set of $n$ independent samples from a $\lambda$-Lipschitz density $p$. Let $\hat{p}$ be the empirical density, $p_\sigma := p \circledast \mathcal{N}(0, \sigma^2)$ and $\hat{p}_\sigma = \hat{p} \circledast \mathcal{N}(0, \sigma^2)$. Then, with probability at least $1 - \delta$ with respect to the sample randomness,*

$$\mathrm{d}_{\mathrm{TV}}(p_\sigma, \hat{p}_\sigma) \lesssim \frac{1}{n} + \frac{1}{\sigma^2 n} + \sqrt{\frac{\log n + \log(1 \vee \lambda) + \log 2/\delta}{\sigma^2 n}}. \tag{4.2}$$

The proof of this result is given in the Appendix.

**Comparing the performance of Algorithms 1 and 2.** Applying Theorem 4.1 directly to the $p_0$ density, we immediately get that the estimate $\hat{p}_t^{(1)}(x)$ obtained by Algorithm 1 satisfies:

$$\mathrm{d}_{\mathrm{TV}}(p_t, \hat{p}_t^{(1)}) \lesssim \frac{1}{n_1} + \frac{1}{\sigma_t^2 n_1} + \sqrt{\frac{\log n_1 + \log(1 \vee \lambda_1) + \log 2/\delta}{\sigma_t^2 n_1}}. \tag{4.3}$$

---

[2]We subtract an $\epsilon$ to allow for approximate mixing of the two distributions and hence smaller annotation times.

[3]This connection has been observed in prior works too, e.g., see [33, 8].

Let us now see what we get by applying Theorem 4.1 to Algorithm 2, which uses samples from the tilted distribution $\tilde{p}_0$. Since this distribution is $\left(\frac{n_1}{n_1+n_2}\lambda_1 + \frac{n_2}{n_1+n_2}\lambda_2\right)$-Lipschitz, we get that:

$$\mathrm{d}_{\mathrm{TV}}(\tilde{p}_t, \hat{p}_t^{(2)}) \lesssim \frac{1}{(n_1+n_2)} + \frac{1}{\sigma_t^2(n_1+n_2)} + \sqrt{\frac{\log(n_1+n_2) + \log(1 \vee \frac{n_1}{n_1+n_2}\lambda_1 + \frac{n_2}{n_1+n_2}\lambda_2) + \log 2/\delta}{\sigma_t^2(n_1+n_2)}},$$

where $\tilde{p}_t := \tilde{p}_0 \circledast \mathcal{N}(0, \sigma_t^2)$.

Further, we have that: $\mathrm{d}_{\mathrm{TV}}(p_t, \hat{p}_t^{(2)}) \leq \mathrm{d}_{\mathrm{TV}}(\tilde{p}_t, p_t) + \mathrm{d}_{\mathrm{TV}}(\tilde{p}_t, \hat{p}_t^{(2)})$. We already have a bound for the second term. To bound the first term, we prove the following theorem.

**Theorem 4.2** (Distance contraction under noise). *Consider distributions $P$ and $Q$ supported on a subset of $\mathbb{R}^d$ with diameter $D$. Then*

$$\mathrm{d}_{\mathrm{TV}}(P \circledast \mathcal{N}(0, \sigma^2 \mathrm{I}), Q \circledast \mathcal{N}(0, \sigma^2 \mathrm{I})) \leq \mathrm{d}_{\mathrm{TV}}(P, Q) \cdot \frac{D}{2\sigma}.$$

Applying this theorem we get that: $\mathrm{d}_{\mathrm{TV}}(\tilde{p}_t, p_t) \leq \frac{1}{2\sigma_t} \mathrm{d}_{\mathrm{TV}}(\tilde{p}_0, p_0) \leq \frac{1}{2\sigma_t} \cdot \frac{n_2}{n_1+n_2} \mathrm{d}_{\mathrm{TV}}(p_0, q_0)$, where for the second inequality we used that $\mathrm{d}_{\mathrm{TV}}(p_0, \tilde{p}_0) \leq \frac{n_2}{n_1+n_2} \mathrm{d}_{\mathrm{TV}}(p_0, q_0)$.

Putting everything together, Algorithm (2) achieves an estimation error:

$$\mathrm{d}_{\mathrm{TV}}(p_t, \hat{p}_t^{(2)}) \lesssim$$
$$\frac{1}{(n_1+n_2)} + \frac{1}{\sigma_t^2(n_1+n_2)} + \sqrt{\frac{\log(n_1+n_2) + \log(1 \vee \frac{n_1}{n_1+n_2}\lambda_1 + \frac{n_2}{n_1+n_2}\lambda_2) + \log 2/\delta}{\sigma_t^2(n_1+n_2)}} + \frac{n_2}{\sigma_t(n_1+n_2)} \mathrm{d}_{\mathrm{TV}}(p_0, q_0).$$

Comparing this with the bound obtained in Equation 4.3, we see that if $n_2$ is sufficiently larger than $n_1$ or if $\lambda_2 \leq \lambda_1$, there is a $t_n^{\min}$ such that for any $t \geq t_n^{\min}$, the upper-bound obtained by Algorithm 2 is better than the upper-bound obtained by Algorithm 1. That implies that for high-diffusion times, using biased data might be helpful for learning, as the bias term (final term) decays with the amount of noise. Going back to equation 4, note that the switching point $t \geq t_n^{\min}$ depends on the distance $\mathrm{d}_{\mathrm{TV}}(\tilde{p}_t, p_t)$ that decays as shown in Theorem 4.2. Once this distance becomes small enough, our computations above suggest that we benefit from biased data. The classifier of Section 3.1, if optimal, exactly tracks the distance $\mathrm{d}_{\mathrm{TV}}(\tilde{p}_t, p_t)$ and, as a result, tracks the switching point.

Table 1: ImageNet results with and without classifier-free guidance.

| **ImageNet-512** | **Train FID ↓** | | | | **Test FID ↓** | | | | **Model Size** | |
| | FID | | FIDv2 | | FID | | FIDv2 | | **Mparams** | **NFE** |
| | no CFG | w/ CFG | no CFG | w/ CFG | no CFG | w/ CFG | no CFG | w/ CFG | | |
| EDM2-XS | **3.57** | 2.91 | **103.39** | 79.94 | 3.77 | 3.68 | 115.16 | 93.86 | 125 | 63 |
| Ambient-o-XS | 3.59 | **2.89** | 107.26 | **79.56** | **3.69** | **3.58** | **115.02** | **92.96** | 125 | 63 |
| EDM2-XXL | 1.91 (1.93) | 1.81 | 42.84 | 33.09 | 2.88 | 2.73 | 56.42 | 46.22 | 1523 | 63 |
| Ambient-o-XXL | 1.99 | 1.87 | 43.38 | 33.34 | 2.81 | 2.68 | 56.40 | 46.02 | 1523 | 63 |
| Ambient-o-XXL+crops | **1.91** | **1.80** | **42.84** | **32.63** | **2.78** | **2.53** | **56.39** | **45.78** | 1523 | 63 |

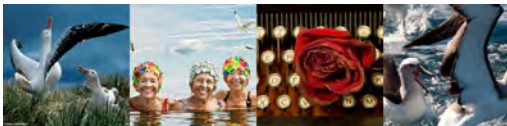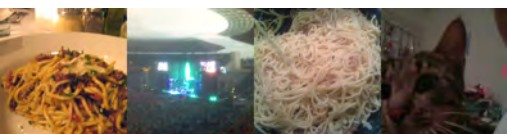

Figure 3: Results using CLIP to obtain the high-quality and the low-quality sets of ImageNet.

## 5 Experiments

**Controlled experiments to show utility from low-quality data.** To verify our method, we first do synthetic experiments on artificially corrupted data. We use EDM [34] as our baseline, and we train networks on CIFAR-10 and FFHQ. For the first experiments, we only use the high-noise part of our Ambient-o method (Section 3.1). We underline that for all of our experiments, we only change the way we use the data, and we keep all the optimization and network hyperparameters as is. We compare against using all the data as equal (despite the corruption) and the filtering strategy of

only training on the clean samples. For evaluation, we measure FID [29] with respect to the full uncorrupted dataset (which is not available during training).

For the blurring experiments, we use a Gaussian kernel with standard deviation $\sigma_B = 0.4, 0.6, 0.8, 1.0$, and we corrupt $90\%$ of the data. We show some corrupted images in Appendix Figure 5a. To obtain annotations, we train a blurry vs clean image classifier under noise, as explained in Section 3.1. For the experiments in the main paper, we use a balanced dataset for the training of the classifier. We ablate the effect of having fewer training samples for the classifier training in Appendix Section E where we show that reducing the number of clean samples available for classifier training leads to a small drop in performance. Once equipped with the trained classifier, each sample is annotated on its own based on the amount of noise that is needed to confuse the classifier (sample dependent annotation). We present our results in Table 2a. As shown, for all corruption strengths, Ambient Omni, significantly outperforms the two baseline methods. In the one to the last column of Table 2a, we further show the average annotation of the classifier. As expected, the average assigned noise level increases as the corruption intensifies.

Table 2: In a controlled experiment with restricted access only to 10% of the clean dataset, our method of Ambient-o uses corrupted and out-of-distribution data to improve performance.

(a) Gaussian blurred data at different levels.

| Method | Parameters Values ($\sigma_B$) | $\bar{\sigma}_{t_n}^{\min}$ | FID |
|---|---|---|---|
| **Only Clean (10%)** | - | - | 8.79 |
| **All data** | 1.0 | 0 | 45.32 |
| | 0.8 | | 28.26 |
| | 0.6 | | 11.42 |
| | 0.4 | | 2.47 |
| **Ambient-o** | 1.0 | 2.84 | **6.16** |
| | 0.8 | 1.93 | **6.00** |
| | 0.6 | 1.38 | **5.34** |
| | 0.4 | 0.22 | **2.44** |

(b) Additional out-of-distribution data.

| Source Data | Additional Data | Method | $\bar{\sigma}_{t_n}^{\max}$ | FID |
|---|---|---|---|---|
| | None | – | – | 12.08 |
| | Cats | Fixed $\sigma$ | 0.2 | 11.14 |
| | Cats | Fixed $\sigma$ | 0.1 | 9.85 |
| Dogs (10%) | Cats | Fixed $\sigma$ | 0.05 | 10.66 |
| | Cats | Fixed $\sigma$ | 0.025 | 12.07 |
| | Cats | Classifier | 0.09 | **8.92** |
| | Procedural | Classifier | 0.042 | 10.98 |
| | None | – | – | 5.20 |
| Cats (10%) | Dogs | Classifier | 0.13 | 5.11 |
| | Wildlife | Classifier | 0.08 | **4.89** |

**Controlled experiments to show utility from out-of-distribution images.** We now want to validate the method developed in Section 3.2 for leveraging out-of-distribution data. To start with, we want to find the mapping between diffusion times and the size of the receptive field required for an optimal denoising prediction. To do so, we take a pre-trained denoising diffusion model and measure the denoising loss at a given location as we increase the size of the context. We provide the corresponding plot in the Supplemental Figures 13, 11. The main finding is that while providing more context always leads to a decrease in the average loss, for sufficiently small noise levels, the loss nearly plateaus before the full image context is provided. That implies that the perfect denoiser for a given noise level only needs to look at a localized part of the image.

Equipped with the mapping between diffusion times and crop sizes, we now proceed to a fun experiment. Specifically, we show that it is possible to use images of cats to improve a generative model for dogs (!) and vice-versa. The cats here represent out-of-distribution data that can be used to improve the performance in the distribution of interest (in our toy example, dogs distribution). To perform this experiment, we train a classifier that discriminates between cats and dog images by looking at crops of various sizes (Section 3.2). Figure 18 shows the predictions of an $8 \times 8$ crops-classifier for an image of a cat, illustrating that there are a number of crops that are misclassified as crops from a dog image. We report results for this experiment in Table 2b and we observe improvements in FID arising from using out-of-distribution data. Beyond natural images, we show that it is even possible to use procedurally generated data from Shaders [6] to (slightly) improve the performance. Figure 19 shows an example of such an image and the corresponding predictions of a crops classifier. Table 2b contains more results and ablations between annotating all the out-of-distribution at a single noise level vs. sample-dependent annotations.

**Corruptions of natural datasets – ImageNet results.** Up to this point, our corrupted data has been artificially constructed to study our method in a controlled setting. However, it turns out that even in real datasets such as ImageNet, there are images with significant degradations such as heavy blur, low lighting, and low contrast, and also images with fantastic detail, clear lightning, and sharp contrast. Here, the high-quality and the low-quality sets are not given and hence we have to estimate them. We opt to use the CLIP-IQA quality metric [66] to separate ImageNet into high-quality (top 10% CLIP-IQA) and low-quality (bottom 90% CLIP-IQA) sets. Figure 3 shows some of the top

and bottom quality images according to our metric. Given the high-quality and low-quality sets, we are now back to the previous setting where we can use the developed Ambient-o methodology. We underline that there is a rich literature regarding quality-assessment methods [69, 68, 49, 67].

We use Ambient-o to refer to our method that uses low-quality data at high diffusion times (Section 5) and Ambient-o+crops to refer to the extended version of our method that uses crops from potentially low-quality images at low-diffusion times. Perhaps surprisingly, there are images in ImageNet that have lower global quality but high-quality crops that we can use for low-noise. We present results in Table 1, where we show the best FID [29] and $FD_{DINOv2}$ obtained by different methods. We show the highest and lowest quality crops, alongside their associated full images, of ImageNet according to CLIP in Figure 15.

As shown in the Table, our method leads to state-of-the-art FID scores, improving over the previous state-of-the-art baseline EDM-2 [35] at both the low and high parameter count settings. The benefits are more pronounced when we measure test FID as our method memorizes significantly less due to the addition of noise during the annotation stage of our pipeline (Section 3.1). Beyond FID, we provide qualitative results in Figure 1 (bottom) and Appendix Figures 7, 8. We further show that the quality of the generated images measured by CLIP increased compared to the baseline in Appendix Table 5. The observed improvements are proof that the ability to learn from data with heterogeneous qualities can be truly impactful for realistic settings beyond synthetic corruptions typically studied in prior work.

**Text-to-image results.** For our final set of experiments, we show how Ambient-o can be used to improve the performance of text-to-image diffusion models. We use the code-base of MicroDiffusion [54], as it is open-data and trainable with modest compute ($\approx$ 2 days on 8-H100 GPUs). Sehwag et al. [54] use four main datasets to train their model: Conceptual Captions (12M) [56], Segment Anything (11M) [41], JourneyDB (4.2M) [63], and DiffusionDB (10.7M) [70]. Of these four, DiffusionDB is of significantly lower quality than the others as it contains solely synthetic data from an outdated diffusion model. This presents an opportunity for the use of our method. Can we use this lower-quality data and improve the performance of the trained network?

We set $\sigma_{\min} = 2$ for all samples from DiffusionDB and $\sigma_{\min} = 0$ for all other datasets and we train a diffusion model with Ambient-o. We note that we did not ablate this hyperparameter and it is quite likely that improved results would be obtained by tuning it or by training a high-quality vs low-quality data classifier for the annotation. Despite that, our trained model achieves a remarkable FID of **10.61** in COCO, significantly improving the baseline FID of 12.37 (Table 4). We present qualitative results in Figure 1 and GPT-4o evaluations on DrawBench and PartiPrompt in Figure 23. Ambient-o and baseline generations for different prompts can be found in Figure 1.

As an additional ablation, we compared our method with the recipe of doing a final fine-tuning on the highest-quality subset, as done in the works of [54, 13]. Compared to this baseline, our method obtained slightly worse COCO FID (10.61 vs 10.27) but obtained much greater diversity, as seen visually in Figure 29 and quantitatively through $> 13\%$ increases in DINO Vendi Diversity on prompts from DiffDB (3.22 vs 3.65.). This corroborates our intuition that data filtration leads to decreased diversity. Ambient-o uses all the data but can strike a fine balance between high-quality and diverse generation.

(a) Measuring fidelity and prompt alignment of generated images on COCO dataset.

| Method | FID-30K ($\downarrow$) | Clip-FD-30K ($\downarrow$) | Clip-score ($\uparrow$) |
|---|---|---|---|
| Baseline | 12.37 | 10.07 | 0.345 |
| Ambient-o | **10.61** | **9.40** | **0.348** |

(b) Measuring performance on the GenEval benchmark.

| Method | Overall | Objects | | Counting | Colors | Position | Color attribution |
| | | Single | Two | | | | |
|---|---|---|---|---|---|---|---|
| Baseline | 0.44 | 0.97 | 0.33 | 0.35 | 0.82 | 0.06 | 0.14 |
| Ambient-o | **0.47** | 0.97 | **0.40** | **0.36** | 0.82 | **0.11** | 0.14 |

Figure 4: Quantitative benefits of Ambient-o on COCO [44] zero-shot generation and GenEval [26].

# 6 Conclusion

Is it possible to get good generative models from bad data? Our framework extracts value from low-quality, synthetic, and out-of-distribution sources. At a time when the ever-growing data demands of GenAI are at odds with the need for quality control, Ambient-o lights a path for both to be achieved simultaneously.

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

## A  Limitations and Future Work

Our work opens several avenues for improvement. On the theoretical side, we aim to establish matching lower bounds to demonstrate that learning from the mixture distribution becomes provably optimal beyond a certain noise threshold. Algorithmically, while our method performs well under high-frequency corruptions, it remains an open question whether more effective training strategies could be used for different types of corruptions (e.g., masking). Moreover, real-world datasets often exhibit patch-wise heterogeneity—for example, facial regions are frequently blurred for privacy, leading to uneven corruption across image crops. We plan to investigate patch-level noise annotations to better capture this structure in future work. Computationally, the full-version of our algorithm requires the training of classifiers for annotations that increases the runtime. This overhead can be avoided by using hand-picked annotation times based on quality proxies as done in our synthetic data experiment. Finally, we believe the true potential of Ambient-o lies in scientific applications, where data often arises from heterogeneous measurement processes.

## B  Theoretical Results

### B.1  Kernel Estimation

**Assumption B.1.** The density $p$ is $\lambda$ lipschitz.

Let $\{X^{(i)}\}_{i=1}^{n}$ a set of $n$ independent samples from a density $p$ that satisfies Assumption B.1. Let $\hat{p}$ be the empirical density on those samples.

We are interested in bounding the total variation distance between $p_\sigma := p \circledast \mathcal{N}(0, \sigma^2)$ and $\hat{p}_\sigma = \hat{p} \circledast \mathcal{N}(0, \sigma^2)$. In particular,

$$\hat{p}_\sigma(x) = \frac{1}{n\sigma} \sum_{i=1}^{n} \phi\left(\frac{X^{(i)} - x}{\sigma}\right), \tag{B.1}$$

where $\phi(u) = \frac{1}{\sqrt{2\pi}} e^{-u^2/2}$ is the Gaussian kernel. We want to argue that the TV distance between $p_\sigma$ and $\hat{p}_\sigma$ is small given sufficiently many samples $n$. For simplicity, let's fix the support of $p$ to be $[0, 1]$. We have:

$$d_{\mathrm{TV}}(p_\sigma, \hat{p}_\sigma) = \frac{1}{2} \int_0^1 |p_\sigma(x) - \hat{p}_\sigma(x)| dx = \sum_{l=0}^{L-1} \int_{l/L}^{(l+1)/L} |p_\sigma(x) - \hat{p}_\sigma(x)| dx \tag{B.2}$$

Now let us look at one of the terms of the summation.

$$\int_{l/L}^{(l+1)/L} |p_\sigma(x) - \hat{p}_\sigma(x)| dx = \int_{l/L}^{(l+1)/L} |p_\sigma(x) - p_\sigma(l/L) + p_\sigma(l/L) - \hat{p}_\sigma(x)| dx \tag{B.3}$$

$$\leq \int_{l/L}^{(l+1)/L} |p_\sigma(x) - p_\sigma(l/L)| dx + \int_{l/L}^{(l+1)/L} |p_\sigma(l/L) - \hat{p}_\sigma(x)| dx. \tag{B.4}$$

We first work on the first term. Using Lemma B.6:

$$\int_{l/L}^{(l+1)/L} |p_\sigma(x) - p_\sigma(l/L)| dx \leq \lambda \int_{l/L}^{(l+1)/L} |x - l/L| dx \tag{B.5}$$

$$= \frac{\lambda}{2L^2}. \tag{B.6}$$

Next, we work on the second term.

$$\int_{l/L}^{(l+1)/L} |p_\sigma(l/L) - \hat{p}_\sigma(x)| dx = \int_{l/L}^{(l+1)/L} |p_\sigma(l/L) - \hat{p}_\sigma(l/L) + \hat{p}_\sigma(l/L) - \hat{p}_\sigma(x)| dx \tag{B.7}$$

$$\leq \int_{l/L}^{(l+1)/L} |p_\sigma(l/L) - \hat{p}_\sigma(l/L)| dx + \int_{l/L}^{(l+1)/L} |\hat{p}_\sigma(l/L) - \hat{p}_\sigma(x)| dx. \tag{B.8}$$

According to Lemma B.5, we have that $\hat{p}_\sigma$ is $\hat{\lambda} = \frac{1}{\sigma^2\sqrt{2\pi e}}$ Lipschitz. Then, the second term becomes:

$$\int_{l/L}^{(l+1)/L} |\hat{p}_\sigma(l/L) - \hat{p}_\sigma(x)|\mathrm{d}x \leq \hat{\lambda}\int_{l/L}^{(l+1)/L} |l/L - x|\mathrm{d}x = \frac{\hat{\lambda}}{2L^2}. \tag{B.9}$$

It remains to bound the following term

$$\int_{l/L}^{(l+1)/L} |p_\sigma(l/L) - \hat{p}_\sigma(l/L)|\mathrm{d}x = \frac{|p_\sigma(l/L) - \hat{p}_\sigma(l/L)|}{L} \tag{B.10}$$

We will be applying Hoeffding's Inequality, stated below:

**Theorem B.2** (Hoeffding's Inequality)**.** *Let $Y_1, ..., Y_n$ be independent random variables in $[a, b]$ with mean $\mu$. Then,*

$$\Pr\left(\left|\frac{1}{n}\sum_{i=1}^n Y_i - \mu\right| \geq t\right) \leq 2\exp\left(-2nt^2/(b-a)^2\right). \tag{B.11}$$

Recall that $\hat{p}_\sigma$ can be written as

$$\hat{p}_\sigma(x) = \frac{1}{n}\sum_{i=1}^n \frac{\phi((X^{(i)} - x)/\sigma)}{\sigma} = \frac{1}{n}\sum_{i=1}^n Y_i, \tag{B.12}$$

in terms of the random variables $Y_i := \frac{\phi((X^{(i)} - x)/\sigma)}{\sigma}$. These random variables are supported in $\left[0, \frac{1}{\sqrt{2\pi\sigma^2}}\right]$. So, for any $x$, we have that:

$$\Pr\left(|\hat{p}_\sigma(x) - \mathbb{E}[\hat{p}_\sigma(x)]| \geq t\right) \leq 2\exp\left(-4\pi\sigma^2 nt^2\right). \tag{B.13}$$

Taking $t = \sqrt{\frac{\log(2L/\delta)}{4\pi\sigma^2 n}}$ and using the above inequality and the union bound, we have that, with probability at least $1 - \delta$, for all $l \in \{0, 1, \ldots, L-1\}$:

$$|\hat{p}_\sigma(l/L) - \mathbb{E}[\hat{p}_\sigma(l/L)]| \leq \sqrt{\frac{\log(2L/\delta)}{4\pi\sigma^2 n}}. \tag{B.14}$$

Let us now compute the expected value of $\hat{p}_\sigma(x)$.

$$\mathbb{E}[\hat{p}_\sigma(x)] = \mathbb{E}\left[\frac{1}{n\sigma}\sum_{i=1}^n \phi\left(\frac{X^{(i)} - x}{\sigma}\right)\right] \tag{B.15}$$

$$= \frac{1}{n\sigma}\sum_{i=1}^n \mathbb{E}\left[\phi\left(\frac{X^{(i)} - x}{\sigma}\right)\right] \tag{B.16}$$

$$= \frac{1}{\sigma}\int p(u)\phi\left(\frac{x - u}{\sigma}\right)\mathrm{d}u \equiv (p \circledast \mathcal{N}(0, \sigma^2))(x) = p_\sigma(x). \tag{B.17}$$

Combining equation B.14 and equation B.17, we get:

$$|\hat{p}_\sigma(l/L) - p_\sigma(x)| \leq \sqrt{\frac{\log(2L/\delta)}{4\pi\sigma^2 n}}. \tag{B.18}$$

Putting everything together we have:

$$\mathrm{d}_{\mathrm{TV}}(p_\sigma, \hat{p}_\sigma) \leq \frac{\lambda}{2L} + \frac{1}{2L\sigma^2\sqrt{2\pi e}} + \sqrt{\frac{\log(2L/\delta)}{4\pi\sigma^2 n}}.$$

Choosing $L = n \cdot \max\{\lambda, 1\}$ we get that:

$$\mathrm{d}_{\mathrm{TV}}(p_\sigma, \hat{p}_\sigma) \lesssim \frac{1}{n} + \frac{1}{\sigma^2 n} + \sqrt{\frac{\log n + \log(1 \vee \lambda) + \log 2/\delta}{\sigma^2 n}}.$$

## B.2 Evolution of parameters under noise

*Proof of theorem 4.2:* We will use the following facts:

*Fact* 1 (Direct corollary of the optimal coupling theorem). There exists a coupling $\gamma$ of $P$ and $Q$, which samples a pair of random variables $(X, Y) \sim \gamma$ such that $\Pr_\gamma[X \neq Y] = \mathrm{d}_{\mathrm{TV}}(P, Q)$.

*Fact* 2. For any $x, y \in \mathbb{R}^d$: $\mathrm{d}_{\mathrm{TV}}(\mathcal{N}(x, \sigma^2 \mathrm{I}), \mathcal{N}(y, \sigma^2 \mathrm{I})) \leq \|x - y\|/2\sigma$

*Proof.* The KL divergence between $\mathcal{N}(\mu_1, \Sigma_1)$ and $\mathcal{N}(\mu_2, \Sigma_2)$ is

$$\mathrm{KL}(\mathcal{N}(\mu_1, \Sigma_1), \mathcal{N}(\mu_2, \Sigma_2)) = \frac{1}{2} \left( \mathrm{tr}(\Sigma_2^{-1} \Sigma_1) + (\mu_2 - \mu_1) \Sigma_2^{-1} (\mu_2 - \mu_1) - d + \log \frac{|\Sigma_2|}{|\Sigma_1|} \right).$$

Applying this general result to our case:

$$\mathrm{KL}(\mathcal{N}(x, \sigma^2 \mathrm{I}), \mathcal{N}(y, \sigma^2 \mathrm{I})) = \frac{1}{2} \left( \frac{\|x - y\|^2}{\sigma^2} \right).$$

We conclude by applying Pinsker's inequality. $\qquad \square$

A corollary of Fact 2 and the optimal coupling theorem is the following:

*Fact* 3. Fix arbitrary $x, y \in \mathbb{R}^d$. There exists a coupling $\gamma_{x,y}$ of $\mathcal{N}(0, \sigma^2 \mathrm{I})$ and $\mathcal{N}(0, \sigma^2 \mathrm{I})$, which samples a pair of random variables $(Z, Z') \sim \gamma_{x,y}$ such that $\Pr_{\gamma_{x,y}}[x + Z \neq y + Z'] = \|x - y\|/2\sigma$.

Now let us denote by $\tilde{P} = P \circledast \mathcal{N}(0, \sigma^2 \mathrm{I})$ and $\tilde{Q} = Q \circledast \mathcal{N}(0, \sigma^2 \mathrm{I})$. To establish our claim in the theorem statement, it suffices to exhibit a coupling $\tilde{\gamma}$ of $\tilde{P}$ and $\tilde{Q}$ which samples a pair of random variables $(\tilde{X}, \tilde{Y}) \sim \tilde{\gamma}$ such that: $\Pr_{\tilde{\gamma}}[\tilde{X} \neq \tilde{Y}] \leq \mathrm{d}_{\mathrm{TV}}(P, Q) \cdot \frac{D}{2\sigma}$. We define coupling $\tilde{\gamma}$ as follows:

1. Sample $(X, Y) \sim \gamma$ (as specified in Fact 1); then

2. sample $(Z, Z') \sim \gamma_{X,Y}$ (as specified in Fact 3); then

3. output $(\tilde{X}, \tilde{Y}) := (X + Z, Y + Z')$.

Let us argue the following:

**Lemma B.3.** *The afore-described sampling procedure $\tilde{\gamma}$ is a valid coupling of $\tilde{P}$ and $\tilde{Q}$.*

*Proof.* We need to establish that the marginals of $\tilde{\gamma}$ are $\tilde{P}$ and $\tilde{Q}$. We will only show that for $(\tilde{X}, \tilde{Y}) \sim \tilde{\gamma}$ according to the afore-described sampling procedure, the marginal distribution of $\tilde{X}$ is $\tilde{P}$, as the proof for $\tilde{Y}$ is identical. Since $\gamma$ is a coupling of $P$ and $Q$, for $(X, Y) \sim \gamma$, the marginal distribution of $X$ is $P$. By Fact 3, conditioning on any value of $X$ and $Y$, the marginal distribution of $Z$ is $\mathcal{N}(0, \sigma^2 \mathrm{I})$. Thus, $\tilde{X} = X + Z$, where $X \sim P$ and independently $Z \sim \mathcal{N}(0, \sigma^2 \mathrm{I})$, and thus the distribution of $\tilde{X}$ is $\tilde{P}$. $\qquad \square$

**Lemma B.4.** *Under the afore-described coupling $\tilde{\gamma}$:* $\Pr_{\tilde{\gamma}}[\tilde{X} \neq \tilde{Y}] \leq \mathrm{d}_{\mathrm{TV}}(P, Q) \cdot \frac{D}{2\sigma}$.

*Proof.* Notice that, when $X = Y$, by Fact 3, $Z = Z'$ with probability 1, and therefore $\tilde{X} = \tilde{Y}$. So for event $\tilde{X} \neq \tilde{Y}$ to happen, it must be that $X \neq Y$ happens and, conditioning on this event, that $X + Z \neq Y + Z'$ happens. By Fact 1, $\Pr_\gamma[X \neq Y] = \mathrm{d}_{\mathrm{TV}}(P, Q)$. By Fact 3, for any realization of $(X, Y)$, $\Pr_{\gamma_{X,Y}}[X + Z \neq Y + Z'] = \frac{\|X - Y\|}{2\sigma} \leq \frac{D}{2\sigma}$, where we used that $P$ and $Q$ are supported on a set with diameter $D$. Putting the above together, the claim follows. $\qquad \square$

$\square$

### B.3 Auxiliary Lemmas

**Lemma B.5** (Lipschitzness of the empirical density). *For a collection of points $X^{(1)}, \dots, X^{(n)}$ consider the function $\hat{p}_\sigma(x) = \frac{1}{n\sigma} \sum_{i=1}^{n} \phi\left(\frac{X^{(i)}-x}{\sigma}\right)$, where $\phi(u) = \frac{1}{\sqrt{2\pi}} e^{-u^2/2}$ is the Gaussian kernel. Then $p_\sigma$ is $\left(\frac{1}{\sigma^2\sqrt{2\pi e}}\right)$-Lipschitz.*

*Proof.* Let us compute the derivative of $\hat{p}_\sigma$:

$$\hat{p}'_\sigma(x) = \frac{1}{n\sigma} \sum_{i=1}^{n} \frac{d}{dx} \phi\left(\frac{x - X^{(i)}}{\sigma}\right) \tag{B.19}$$

$$= \frac{1}{\sqrt{2\pi}n\sigma} \sum_{i=1}^{n} \exp\left(-(X^{(i)} - x)^2/(2\sigma^2)\right) \frac{X^{(i)} - x}{\sigma^2} \tag{B.20}$$

$$\leq \frac{1}{\sqrt{2\pi}\sigma^2} \max_u \exp(-u^2/2)u \tag{B.21}$$

$$\leq \frac{1}{\sigma^2\sqrt{2\pi e}}. \tag{B.22}$$

$\square$

**Lemma B.6** (Lipschitzness of a density convolved with a Gaussian). *Let $p$ be a density that is $\lambda$-Lipschitz. Let $p_\sigma = p \circledast \mathcal{N}(0, \sigma^2 I)$. Then, $p_\sigma$ is also $\lambda$-Lipschitz.*

*Proof.* Let us denote with $\phi_\sigma(\cdot)$ the Gaussian density with variance $\sigma^2$. We have that:

$$p_\sigma(x) - p_\sigma(y) = \int (p(x - \tau) - p(y - \tau))\phi_\sigma(\tau)d\tau \Rightarrow \tag{B.23}$$

$$|p_\sigma(x) - p_\sigma(y)| \leq \int |p(x - \tau) - p(y - \tau)|\phi_\sigma(\tau)d\tau \tag{B.24}$$

$$\leq \lambda|x - y| \cdot \int \phi_\sigma(\tau)d\tau \tag{B.25}$$

$$= \lambda|x - y|. \tag{B.26}$$

$\square$

## C  Additional Results

### C.1  CIFAR-10 controlled corruptions

Figures 5a, 5b and 6 show gaussian blur, motion blur, and JPEG corrupted CIFAR-10 images respectively at different levels of severity. Appendix Table 3 shows results for JPEG compressed data at different levels of compression. We also tested our method for motion blurred data with high severity, visualized in the last row of Appendix Figure 6), obtaining a best FID of 5.85 (compared to 8.79 of training on only the clean data).

Table 3: Results for learning from JPEG compressed data on CIFAR-10.

| Method | Dataset | Clean (%) | Corrupted (%) | JPEG Compression (Q) | $\bar{\sigma}_{t_n}^{\min}$ | FID |
|---|---|---|---|---|---|---|
| **Only Clean** | Cifar-10 | 10 | 0 | – | – | 8.79 |
| **Ambient Omni** | Cifar-10 | 10 | 90 | 15% | 1.60 | 6.67 |
| | | | | 18% | 1.40 | 6.43 |
| | | | | 25% | 1.27 | 6.34 |
| | | | | 50% | 1.03 | 5.94 |
| | | | | 75% | 0.81 | 5.57 |
| | | | | 90% | 0.63 | 4.72 |

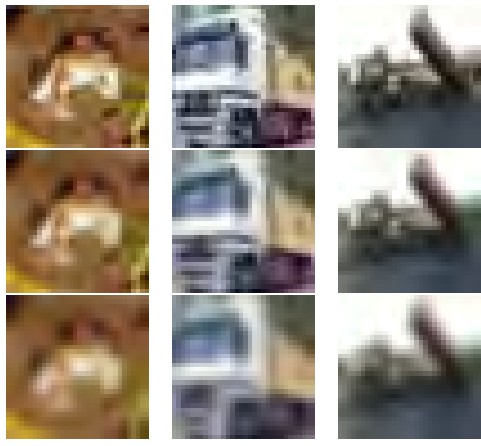
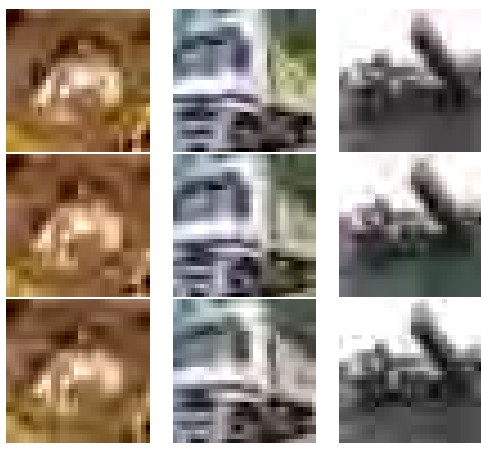

(a) CIFAR-10 images corrupted with blur at increasing levels ($\sigma_B = 0.4, 0.6, 1.0$).

(b) CIFAR-10 images corrupted with JPEG at compression rates: $25\%, 18\%, 15\%$ respectively.

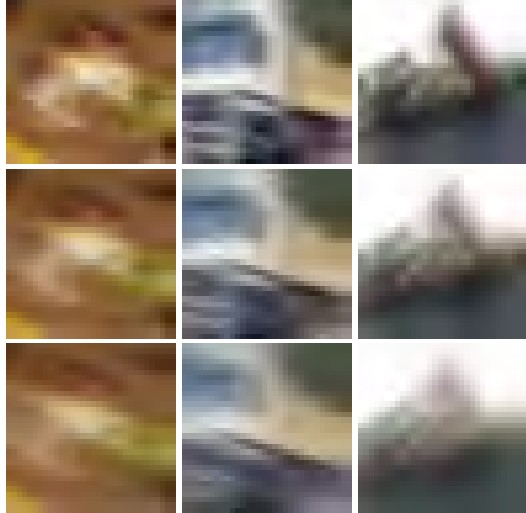

Figure 6: CIFAR-10 images corrupted with motion blur at increasing levels of corruption.

## C.2 FFHQ-64x64 controlled corruptions

In Appendix 4 we show additional results for learning from blurred data on the FFHQ dataset. Similarly to the main paper, we observe that our Ambient-o algorithm leads to improvements over just using the high-quality data that are inversely proportional to the corruption level.

Table 4: Results for learning from blurred data, FFHQ.

| Method | Dataset | Clean (%) | Corrupted (%) | Parameters Values ($\sigma_B$) | $\bar{\sigma}_{t_n}^{\min}$ | FID |
|---|---|---|---|---|---|---|
| **Only Clean** | FFHQ | 10 | 0 | - | - | 5.12 |
| **Ambient Omni** | FFHQ | 10 | 90 | 0.8 | 2.89 | 4.95 |
| | | 10 | 90 | 0.6 | 2.12 | 4.65 |
| | | 10 | 90 | 0.4 | 0.63 | 3.32 |

 ## C.3  ImageNet results

 In the main paper, we used FID as a way to measure the quality of generated images. However, FID
 is computed with respect to the test dataset that might also have samples of poor quality. Further,
 during FID computation, quality and diversity are entangled. To disentangle the two, we generate
 images using the EDM-2 baseline and our Ambient-o model and we use CLIP to evaluate the quality
 of the generated image (through the CLIP-IQA metric implemented in the PIQ package [38, 37]). We
 present results and win-rates in Table 5. As shown, Ambient-o achieves a better per-image quality
 compared to the baseline despite using exactly the same model, hyperparameters, and optimization
 algorithm. The difference comes solely from better use of the available data.

Table 5: Additional comparison between EDM-2 XXL and our Ambient-o model using the CLIP
IQA metric for image quality assesment. Ambient-o leads to improved scores despite using the exact
same architecture, data and hyperparameters. For this experiment, we use the models with guidance
optimized for DINO FD since they are the ones producing the higher quality images.

| Metric | EDM-2 [35] XXL | Ambient-o XXL crops |
|---|---|---|
| Average CLIP IQA score | 0.69 | **0.71** |
| Median CLIP IQA score | 0.79 | **0.80** |
| Win-rate | $47.98\%$ | $\mathbf{52.02}\%$ |

 # D  Ambient diffusion implementation details and loss ablations

 Similar to the EDM-2 [35] paper, we use a pre-condition weight to balance the importance of different
 diffusion times. Specifically, we modulate the EDM2 weight $\lambda(\sigma)$ by a factor:

$$\lambda_{\text{amb}}(\sigma, \sigma_{\min}) = \sigma^4/(\sigma^2 - \sigma_{\min}^2)^2 \tag{D.1}$$

 for our ambient loss based on a similar analysis to [35]. We further use a buffer zone around the
 annotation time of each sample to ensure that the loss doesn't have singularities due to divisions by 0.
 We ablate the precondition term and the buffer size in Appendix Table 6.

Table 6: Ablation study of ambient weight and stability buffer on Cifar-10 with 10% clean data and
90% corrupted data with blur of 0.6.

| Method | FID ↓ |
|---|---|
| *No ambient preconditioning weight and no buffer:* | |
| $\lambda_{\text{amb}}(\sigma, \sigma_{\min}) = 1$ & $\sigma > \sigma_{\min}$ | 5.49 |
| *Adding ambient preconditioning weight:* | |
| + Weight $\lambda_{\text{amb}}(\sigma, \sigma_{\min}) = \sigma^4/(\sigma^2 - \sigma_{\min}^2)^2$ | 5.36 |
| *Adding stability buffer/clipping:* | |
| + Clip $\lambda_{\text{amb}}(\sigma, \sigma_{\min})$ at 2.0 | 5.35 |
| + Clip $\lambda_{\text{amb}}(\sigma, \sigma_{\min})$ at 4.0 | 5.69 |
| + Buffer $\lambda_{\text{amb}}(\sigma, \sigma_{\min})$ at 2.0 i.e. $\sigma > \sqrt{2}\sigma_{\min}$ | 5.40 |
| + Buffer $\lambda_{\text{amb}}(\sigma, \sigma_{\min})$ at 4.0 i.e. $\sigma > (2/\sqrt{3})\sigma_{\min}$ | **5.34** |

 For our ablations, we focus on the setting of training with $10\%$ clean data and $90\%$ corrupted data
 with Gaussian blur of $\sigma_B = 0.6$. Using no ambient pre-conditioning and no buffer, we obtain an
 FID of 5.56. In the same setting, adding the ambient pre-conditioning weight $\lambda_{\text{amb}}(\sigma, \sigma_{\min})$ improves
 FID by 0.13 points. Next, we ablate two strategies to mitigate the impact of the singularity of
 $\lambda_{\text{amb}}(\sigma, \sigma_{\min})$ at $\sigma = \sigma_{\min}$. The first strategy clips the ambient pre-conditioning weight at a specified
 maximum value $\lambda_{\text{amb}}^{\text{MAX}}$, but still trains for $\sigma$ arbitrarily close to $\sigma_{\min}$. The second strategy also specifies
 a maximum value, but imposes a buffer

$$\sigma > \sqrt{1 + \frac{1}{\lambda_{\text{amb}}^{\text{MAX}} - 1}} \sigma_{\min} \tag{D.2}$$

that restricts training to noise levels $\sigma$ such that $\lambda_{\mathrm{amb}}(\sigma, \sigma_{\min}) \leq \lambda_{\mathrm{amb}}^{\mathrm{MAX}}$. Clipping the ambient weight to $\lambda_{\mathrm{amb}}^{\mathrm{MAX}} = 2.0$ minimally improves FID to 5.35, but clipping to 4.0 significantly worsens it to 5.69. Adding a buffer at $\lambda_{\mathrm{amb}}^{\mathrm{MAX}} = 2.0$ slightly worsens FID to 5.40, but slackening the buffer to 4.0 minimally improves FID to 5.34. We opt for the buffering strategy in favor of the clipping strategy since performance appears convex in the buffer parameter, and because it obtains the best FID.

# E   Annotation ablations

We ablate the choice of using a fixed annotation vs sample-adaptive annotations in Appendix Table 7. We find that using sample-adaptive annotations achieves improved results. Nevertheless, both annotation methods yield improvements over the training on filtered data and the training on everything baselines. To show that our method works for more corruption types, we perform an equivalent experiment with JPEG compressed data at different compression ratios and we achieve similar results, presented in Appendix Table 3. We ablate the impact of the amount of training data and the number of training iterations on the classifier annotations in Appendix Section E. We show results for motion blur (Figure 6 and Section C.1) and for the FFHQ dataset (Table 4).

**Balanced vs unbalanced data:** We ablate the impact of classifier training data on the setting of CIFAR-10 with 10% clean data and 90% corrupted data with gaussian blur with $\sigma_B = 0.6$. When annotating with a classifier trained on the same unbalanced dataset we train the diffusion model on we obtained a best FID of 6.04, compared to the 5.34 obtained if we train on a balanced dataset.

**Training iterations:** We ablate the impact of classifier training iterations on the setting of CIFAR-10 with 10% clean data and 90% corrupted data with JPEG compression at compression rate of 18%, training the classifier with a balanced dataset. We report minute variations in the best FID, obtaining 6.50, 6.58, and 6.49 when training the classifier for 5e6, 10e6, and 15e6 images worth of training respectively.

Table 7: Comparison with baselines for training with data corrupted by Gaussian Blur at different levels. The dataset used in this experiment is CIFAR-10.

| Method | Clean (%) | Corrupted (%) | Parameters Values ($\sigma_B$) | $\bar{\sigma}_{t_n}^{\min}$ | FID |
|---|---|---|---|---|---|
| **Only Clean** | 10 | 0 | - | - | 8.79 |
| **No annotations** | 10 | 90 | 1.0 | 0 | 45.32 |
|  |  |  | 0.8 |  | 28.26 |
|  |  |  | 0.4 |  | 2.47 |
| **Single annotation** | 10 | 90 | 1.0 | 2.32 | 6.95 |
|  |  |  | 0.8 | 1.89 | 6.66 |
|  |  |  | 0.4 | 0.00 | 2.47 |
| **Classifier annotations** | 10 | 90 | 1.0 | 2.84 | 6.16 |
|  | 10 | 90 | 0.8 | 1.93 | 6.00 |
|  | 10 | 90 | 0.4 | 0.22 | 2.44 |

# F   Training Details

## F.1   Formation of the high-quality and low-quality sets.

In the theoretical problem setting we assumed the existence of a good set $S_G$ from the clean distribution and a bad set $S_B$ from the corrupted distribution. In practice, we do not actually possess these sets initially, but we can construct them so long as we have access to a measure of "quality". Given a function on images which tells us wether its good enough to generate or not e.g. CLIP-IQA quality [66] greater than some threshold, we can define our good set $S_G$ as the good enough images and $S_B$ as the complement. From this point on we can apply the methodology of ambient-o as developed, either employing classifier annotations as in our pixel diffusion experiments, or fixed annotations as in our large scale ImageNet and text-to-image experiments.

### F.2 Datasets

**CIFAR-10.** CIFAR-10 [42] consists of 60,000 32x32 images of ten classes (airplane, automobile, bird, cat, deer, dog, frog, horse, ship, and truck).

**FFHQ.** FFHQ [36] consists of 70,000 512x512 images of faces from Flickr. We used the dataset at 64x64 resolution for our experiments.

**AFHQ.** AFHQ [12] consists of 5,653 images of cats, 5,239 images of dogs and 5,000 images of wildlife, for a total of 15,892 images.

**ImageNet.** ImageNet [20] consists of 1,281,167 images of variable resolution from 1000 classes.

**Conceptual Captions.** Conceptual Captions [56] consists of 12M (image url, caption) pairs.

**Segment Anything.** Segment Anything [41] consists of 11.1M high-resolution images annotated with segmentation masks. Since the original dataset did not have real captions, we use the same LLaVA generated captions created by the MicroDiffusion [54] paper.

**JourneyDB.** JourneyDB consists of 4.4M synthetic image-caption pairs from Midjourney [63].

**DiffusionDB.** DiffusionDB consists of 14M synthetic image-caption pairs, mostly generated from Stable Diffusion models [70]. We use the same 10.7M quality-filtered subset created by the MicroDiffusion paper [54].

### F.3 Diffusion model training

**CIFAR-10.** We use the EDM [34] codebase as a reference to train class-conditional diffusion models on CIFAR-10. The architecture is a Diffusion U-Net [60] with ~55M paramemeters. We use the Adam optimizer [40] with learning rate 0.001, batch size 512, and no weight decay. While the original EDM paper trained for $200 \times 10^6$ images worth of training, when training with corrupted data we saw best results around $20 \times 10^6$ images. On a single 8xV100 node we achieved a throughput of 0.8s per 1k images, for an average of 4.4h per training run.

**FFHQ.** Same as for CIFAR-10, except learning was set to $2e - 4$, we trained for a maximum of $100 \times 10^6$ images worth of training, and saw best results around $30 \times 10^6$ images worth.

**AFHQ.** Same as FFHQ.

**ImageNet.** We use the EDM2 [35] codebase as a reference to train class-conditional diffusion models on ImageNet. The architecture is a Diffusion U-Net [60] with ~125M paramemeters. We use the Adam optimizer [40] with reference learning rate 0.012, batch size 2048, and no weight decay. Same as the original codebase, we trained for ~2B worth of images. On 32 H200 GPUs, XS models took ~3 days to train, while XXL models took ~7 days.

**MicroDiffusion.** We use the MicroDiffusion codebase [54] as a reference to train text-to-image models on an academic budget. We follow their recipe exactly, changing only the standard denoising diffusion loss to the ambient diffusion loss. The architecture is a Diffusion Transformer [50] utilizing Mixture-of-Experiments (MoE) feedforward layers [57, 31], with ~1.1B paramemeters. We use the AdamW optimizer [40] with reference learning rates $2.4e - 4/8e - 5/8e - 5/8e - 5$ for each of the four phases and batch size 2048 for all phases. On 8 H200 GPUs, training takes ~2 days to train.

### F.4 Classifier training

Classifier training is done using the same optimization recipe (optimizer, learning rate, batch size, etc.) as diffusion model training, except we change the architecture to an encoder-only "Half-Unet", simply by removing the decoder half of the original UNet architecture. The training of the classifier is substantially shorter compared to the diffusion training since classification is task is easier than generation.

 # G   Additional Figures

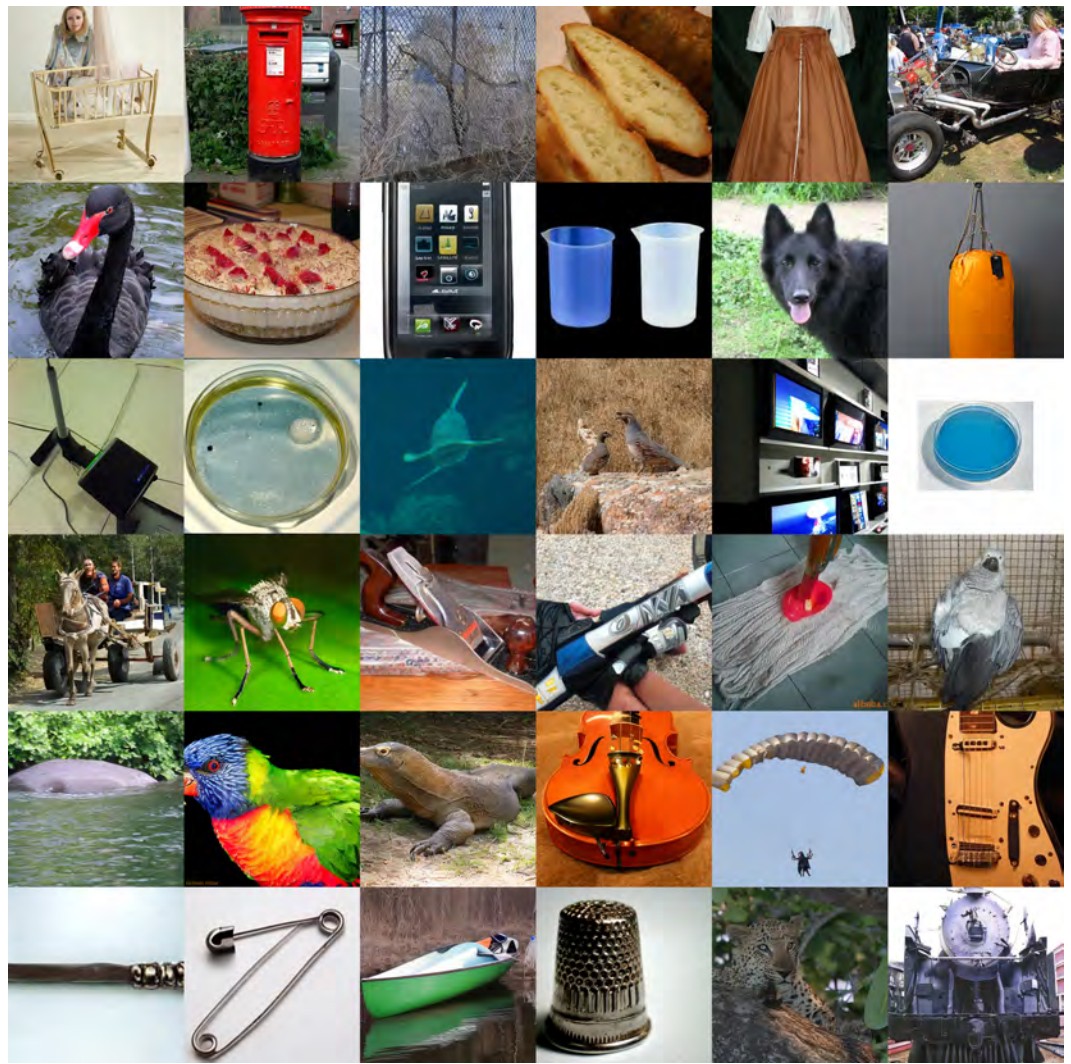

Figure 7: Uncurated generations from our Ambient-o  XXL model trained on ImageNet.

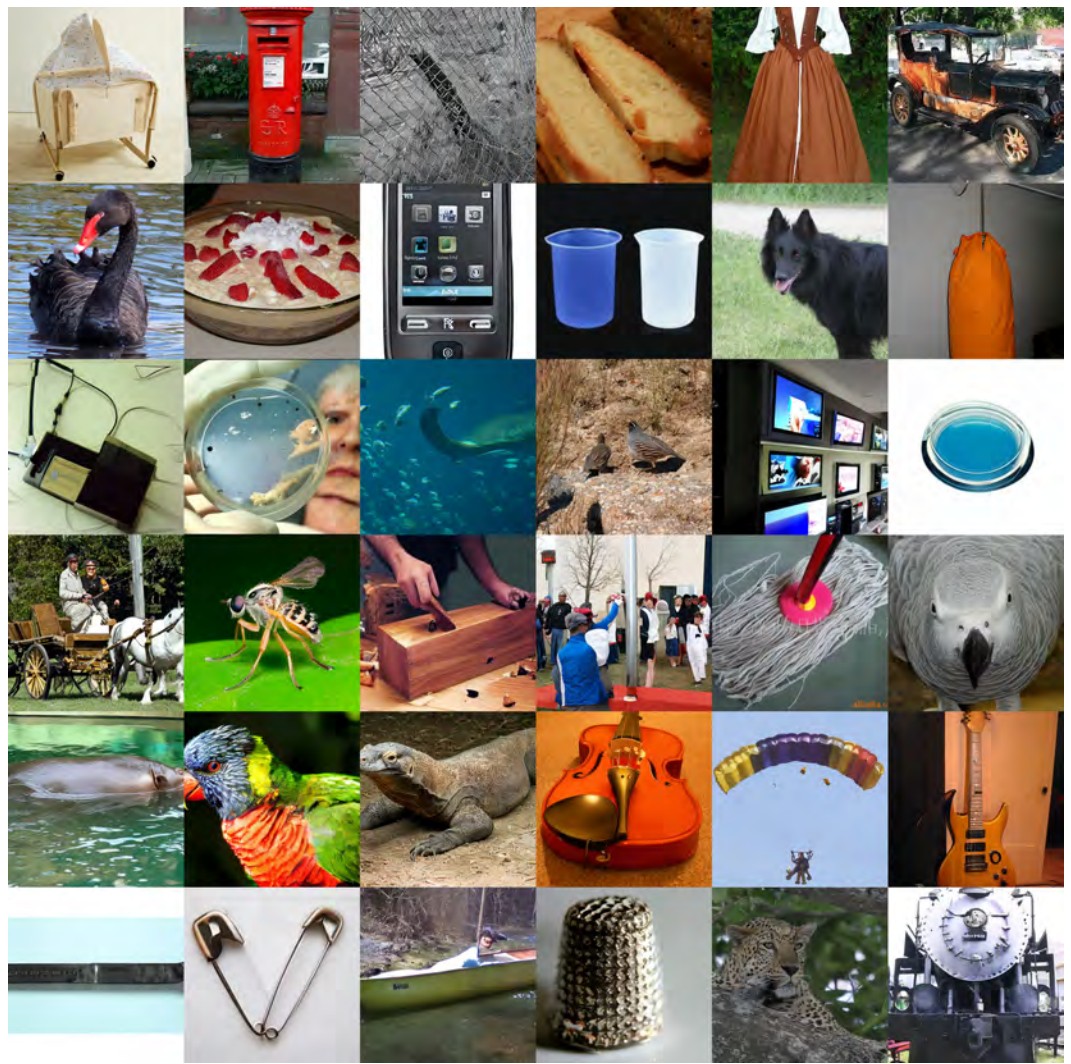

Figure 8: Uncurated generations from our Ambient-o+crops XXL model trained on ImageNet.

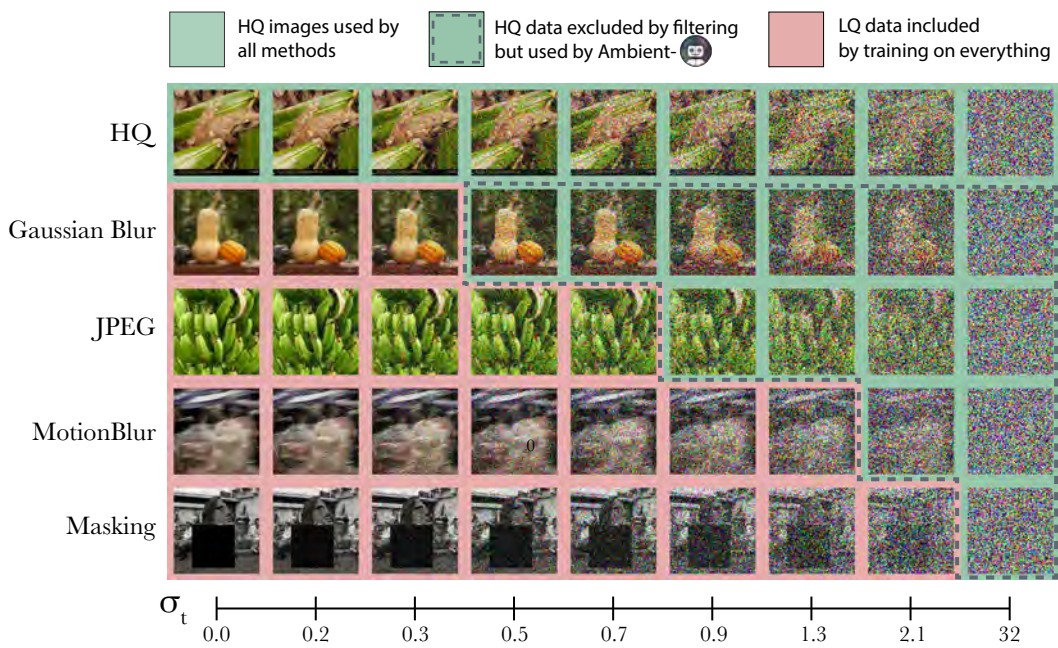

Figure 9: **Visual summary of our method for using low-quality data at high-noise.** We see how the various corrupted images become indistinguishable from the High Quality (HQ) after a minimum noise level. These noisy versions of Low Quality (LQ) images are actually high-quality data, which filtering approaches discard, but Ambient Omni uses.

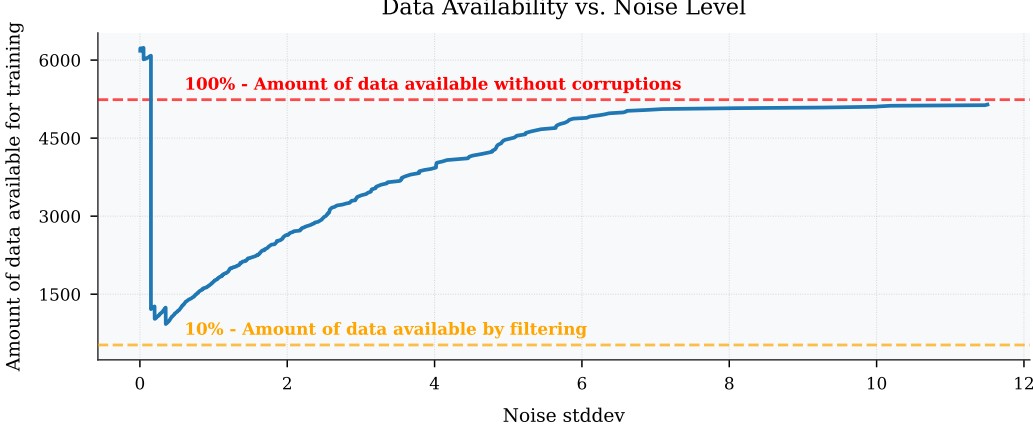

Figure 10: Amount of samples available at each noise level when training a generative model for dogs in the following setting: (1) we have 10% of the dogs dataset uncorrupted, (2) we have the other 90% of the dogs dataset corrupted with gaussian blur with $\sigma_B = 0.6$, and (3) we have 100% of the clean dataset of cats. At low noise levels, we can train on both the high quality dogs and a lot of the cats, resulting in $> 100\%$ of samples available relative to the original dogs dataset size. As the noise level starts to increase, we stop being able to use to the out-of-distribution cat samples, but start gaining some blurry dog samples. As the noise level approaches the maximum all the blurry dogs become available for training, such that the amount of data available approaches 100%.

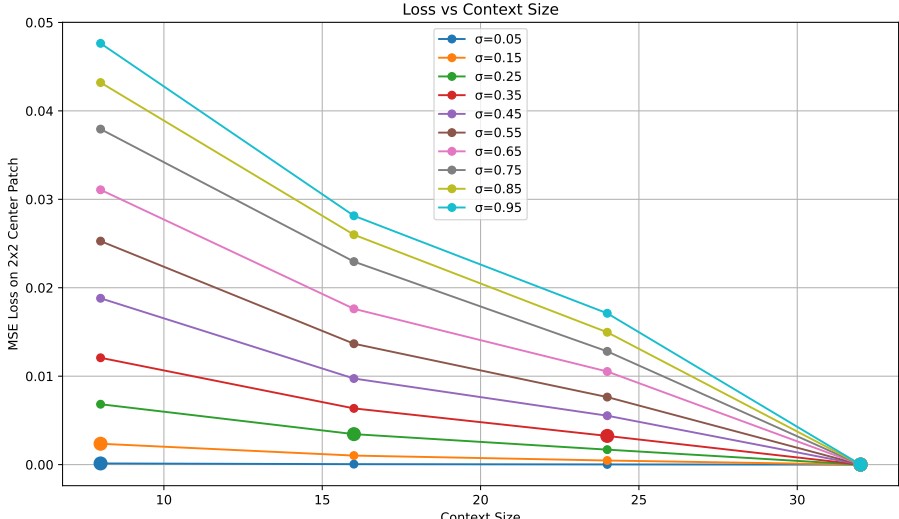

Figure 11: ImageNet-512x512: denoising loss of an optimally trained model, measured at $2 \times 2$ center patch, as we increase the context size given to the model (horizontal axis) and the noise level (different curves). As expected, for higher noise, more context is needed for optimal denoising. The large dot on each curve marks the point where the loss nearly plateaus.

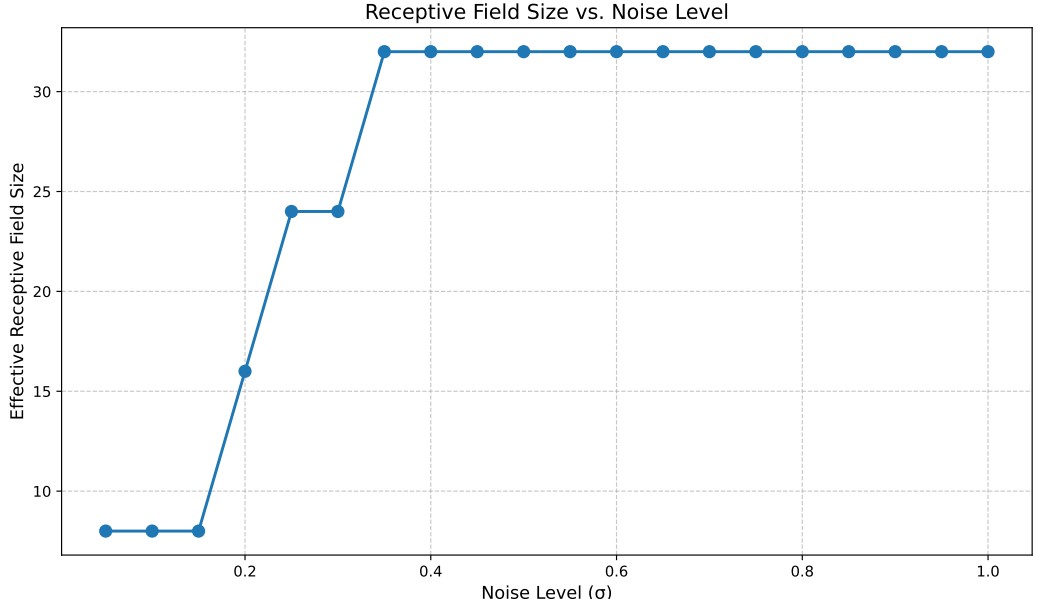

Figure 12: ImageNet-512x512: context size needed to be within $\epsilon = 1e - 3$ of the optimal loss for different noise levels. As expected, for higher noise, more context is needed for optimal denoising.

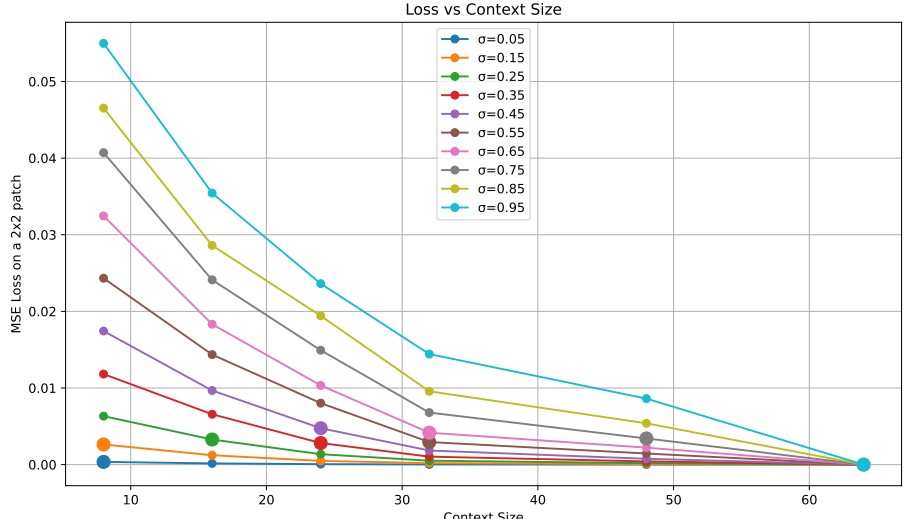

Figure 13: FFHQ: denoising loss of an optimally trained model, measured at $2 \times 2$ center patch, as we increase the context size given to the model (horizontal axis) and the noise level (different curves). As expected, for higher noise, more context is needed for optimal denoising. The large dot on each curve marks the point where the loss nearly plateaus.

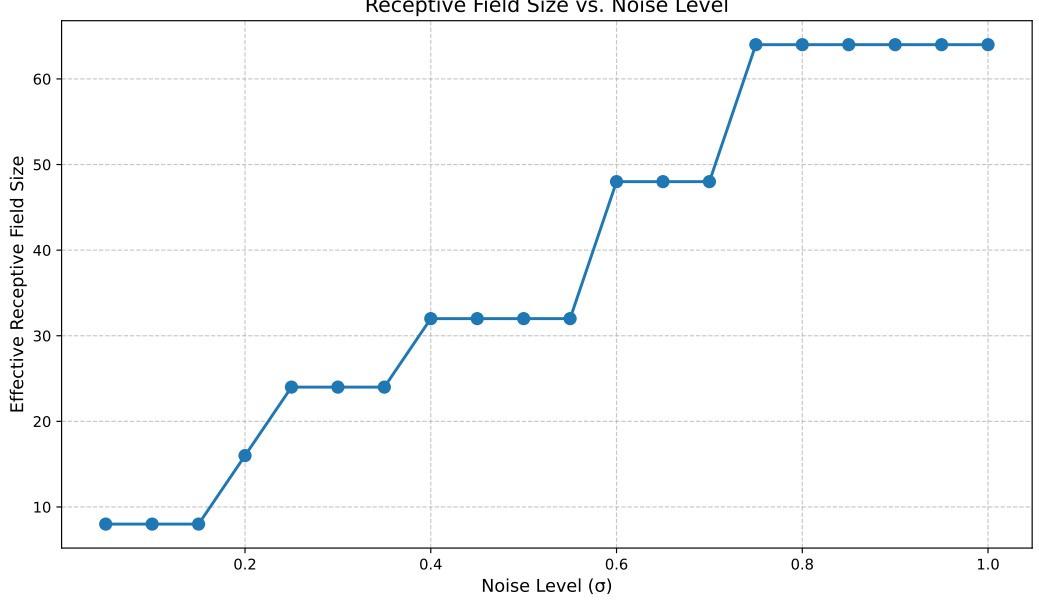

Figure 14: FFHQ: context size needed to be within $\epsilon = 1e - 3$ of the optimal loss for different noise levels. As expected, for higher noise, more context is needed for optimal denoising.

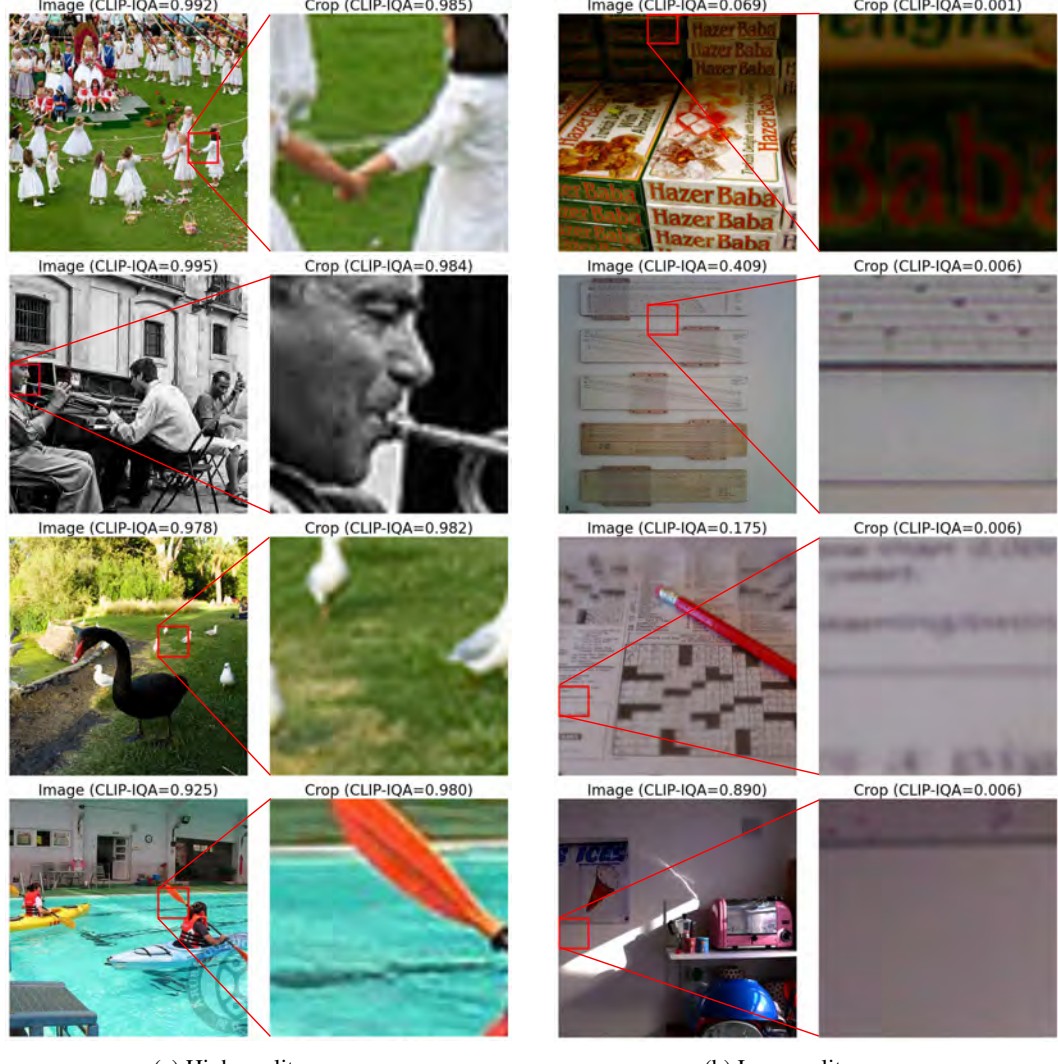

(a) High quality crops

(b) Low quality crops

Figure 15: Results using CLIP to find (a) high-quality and (b) low-quality crops on ImageNet.

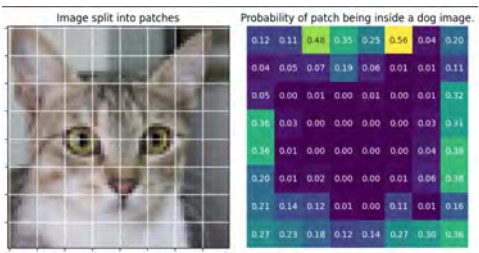

(a) Cat image and classification probabilities over patches.

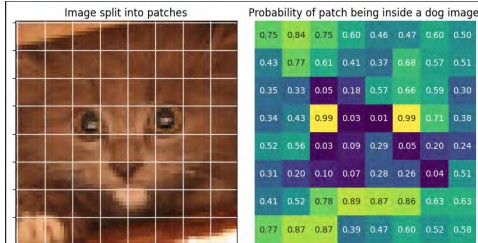

(b) Cat image and classification probabilities over patches.

Figure 16: Two examples of cats from the AFHQ dataset. We partition each cat into non overlapping patches and we compute the probabilities of the patch belonging to an image of a dog using a cats vs dogs classifier trained on patches. The cat on the right has a lot more patches that could belong to a dog image according to the classifier, possibly due to the color or the texture of the fur.

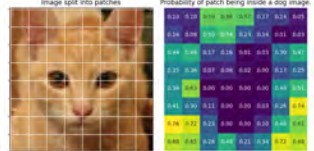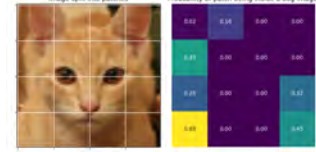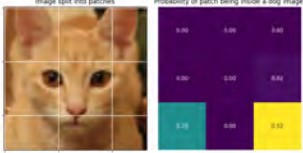

(a) Cat annotated by a cats vs. dogs classifier that operates with crops of size 8.

(b) Cat annotated by a cats vs. dogs classifier that operates with crops of size 16.

(c) Cat annotated by a cats vs. dogs classifier that operates with crops of size 24.

Figure 17: Patch-based annotations of a cat image from AFHQ using cats vs. dogs classifiers trained on different patch sizes.

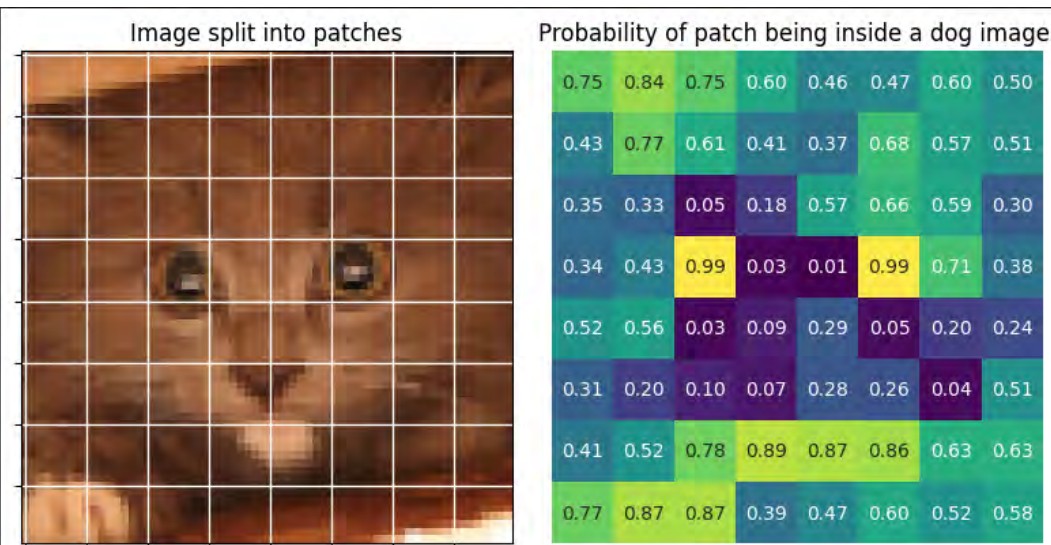

Figure 18: Patch level probabilities for dogness in a cat image.

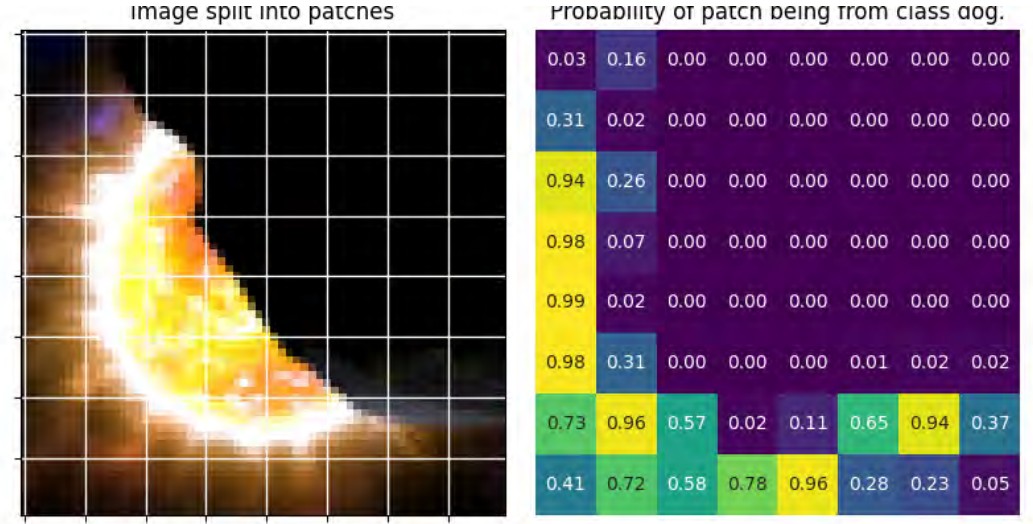

Figure 19: Patch level probabilities for dogness in a synthetic image (procedural program). The cat has more useful patches than this non-realistic procedural program.

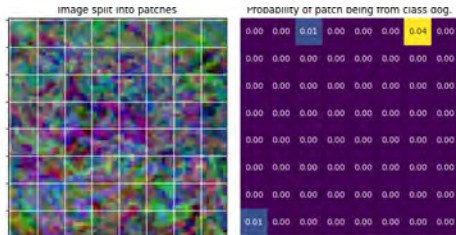
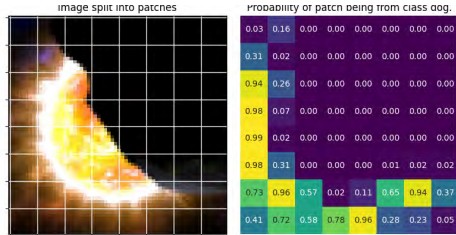

(a) Synthetic image and classification probabilities over patches.

(b) Synthetic image and classification probabilities over patches.

Figure 20: Two examples of procedurally generated images. We partition each image into non overlapping patches and we compute the probabilities of the patch belonging to an image of a dog using a synthetic image vs dogs classifier trained on patches. The image on the right has a lot more patches that could belong to a dog image according to the classifier, possibly due to the color or the texture.

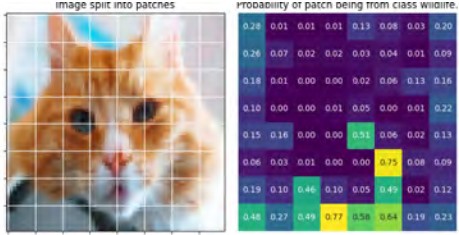
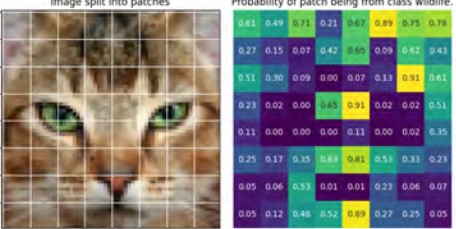

(a) Cat image and classification probabilities over patches.

(b) Cat image and classification probabilities over patches.

Figure 21: Two examples of cat images. We partition each image into nonoverlapping patches and we compute the probabilities of the patch belonging to an image of wildlife using a cats vs wildlife classifier trained on patches. The image on the right has a lot more patches that could belong to a wildlife image according to the classifier, possibly due to the color or the texture.

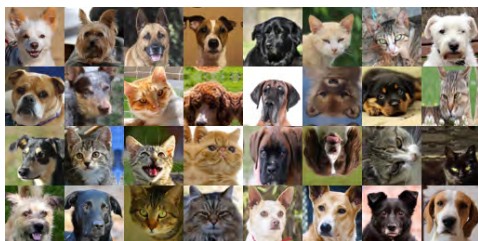
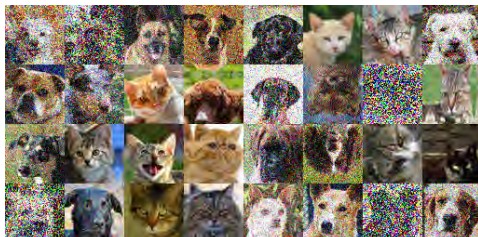

(a) Example batch.

(b) Noisy batch.

Figure 22: Example batch.

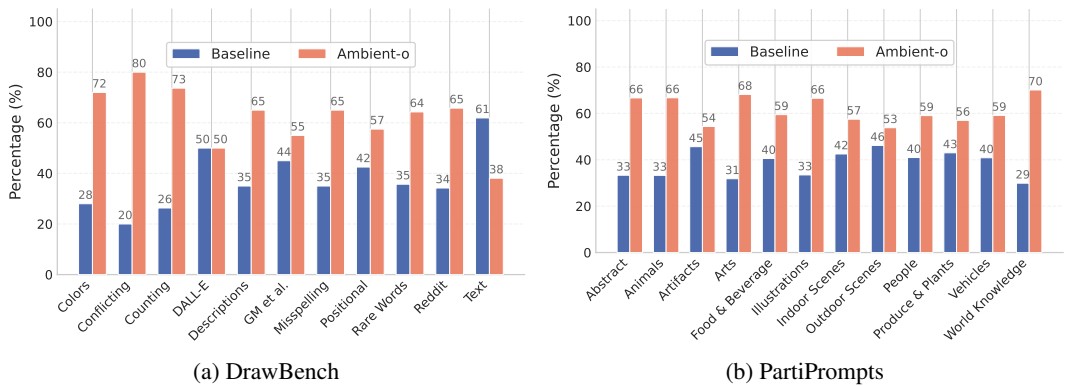

(a) DrawBench

(b) PartiPrompts

Figure 23: Assessing image quality with GPT-4o on DrawBench and PartiPrompts.

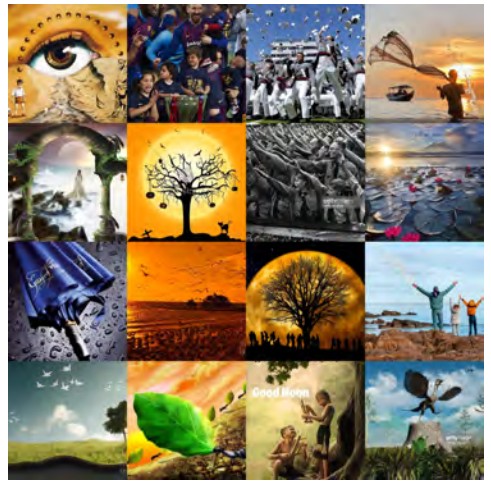

(a) Highest quality images from CC12M according to CLIP.

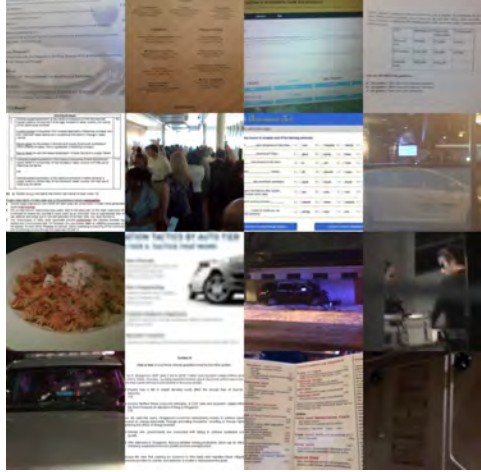

(b) Lowest quality images from CC12M according to CLIP.

Figure 24: CLIP annotations for quality of images from CC12M.

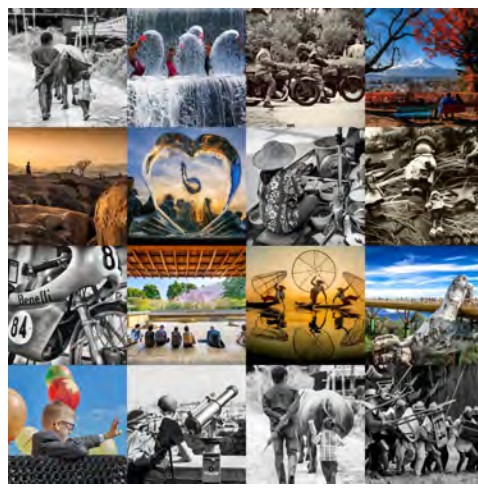

(a) Highest quality images from SA1B according to CLIP.

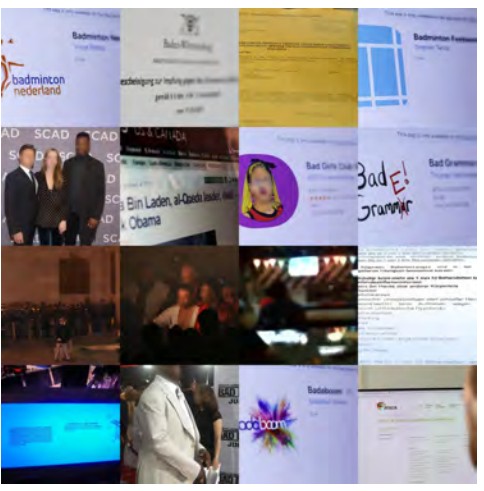

(b) Lowest quality images from SA1B according to CLIP.

Figure 25: CLIP annotations for quality of images from SA1B.

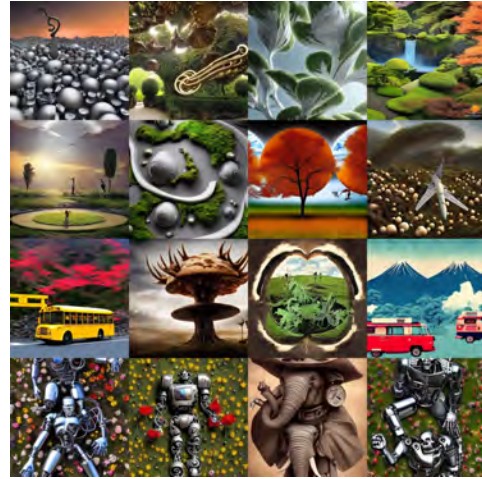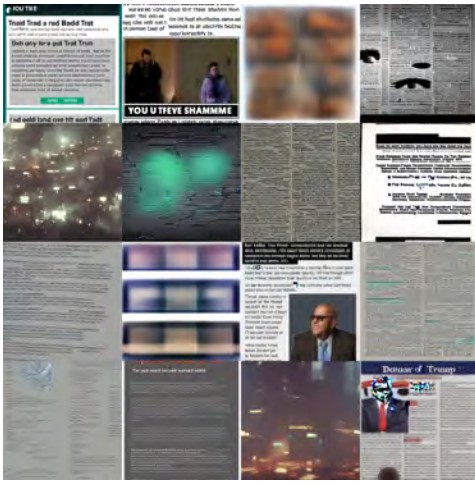

(a) Highest quality images from DiffDB according to CLIP.

(b) Lowest quality images from DiffDB according to CLIP.

Figure 26: CLIP annotations for quality of images from DiffDB.

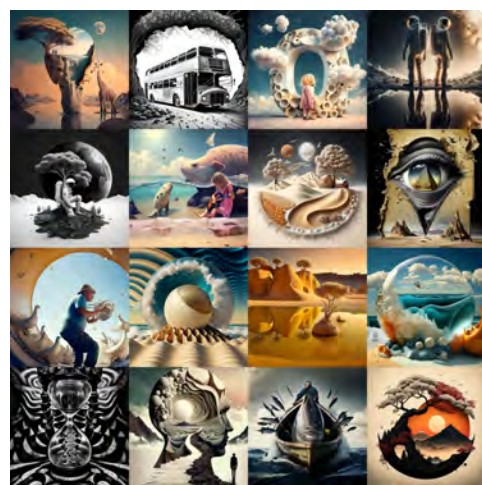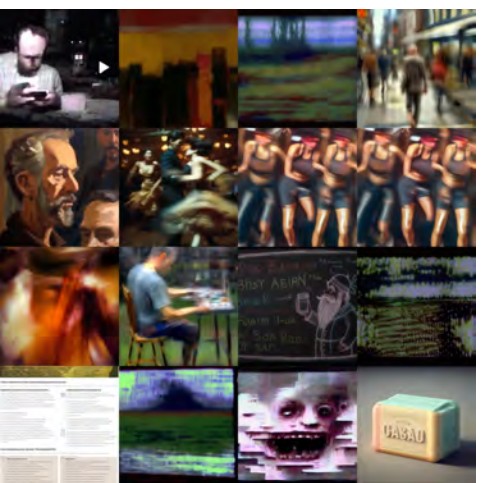

(a) Highest quality images from JDB according to CLIP.

(b) Lowest quality images from JDB according to CLIP.

Figure 27: CLIP annotations for quality of images from JDB.

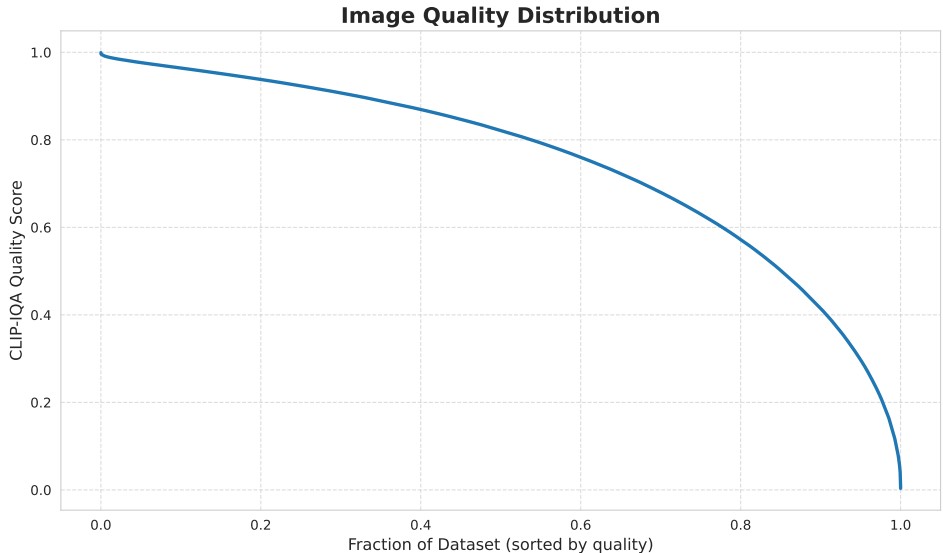

Figure 28: Distribution of image qualities according to CLIP for ImageNet-512.

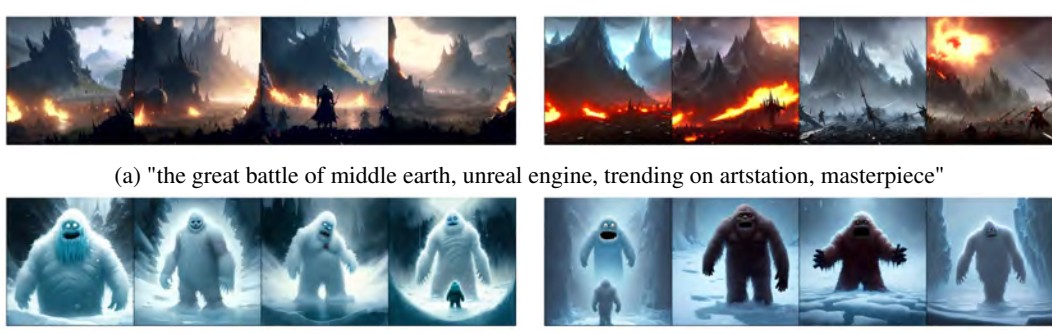

(a) "the great battle of middle earth, unreal engine, trending on artstation, masterpiece"

(b) "an abominable snowman trapped in ice by greg rutkowski"

Figure 29: **Examples of mode collapse**. Left: baseline model finetuned on a high-quality subset. Right: Ambient-o model using all the data. As shown, finetuning decreases output diversity.

