# OpenReview forum: "Ambient Diffusion Omni"
_NeurIPS.cc/2025/Workshop/Reliable_ML — NeurIPS 2025 - Reliable ML Workshop_

### Official Review · Reviewer_uf3s · 2025-09-19
**The paper was a fun read and the experiments seemed thoughtfully designed and interesting. I had some questions about the theory.**

**Rating:** 9
**Confidence:** 3

**Review:**

Summary: Ambient Diffusion Omni proposes a method and framework for generative modeling in the regime where one has access to a (relatively small) amount of “clean” data in addition to arbitrarily corrupted data. This work generalizes prior work (which usually assumes to know the degradation process) by allowing arbitrary corruption. The framework uses the fact that at high levels of noise, the corrupted data is close to data corrupted with additive Gaussian noise, but at the cost of losing information by only using samples from D for diffusion times  $t \geq t_n^{min}$. The paper includes theoretical insight for why using additional corrupted samples could help the learning task (Section 4) and includes several ablations and experimental studies to explore various aspects of this method.

Strengths:
The paper was a fun read and had strong ideas, theoretical justification, experiments, and presentation.
- The concept of being able to use other arbitrarily corrupted data in addition to those from the clean distribution to generate $p_0$ is interesting and practical. It is also interesting that the method appears to work even when $|S_G| << |D|$. The experiment which improves a generative model for dogs using cats was interesting and fun.
- The experiments in both the controlled setting and the natural datasets, and text-to-image results are well-designed. The discussions and ablations done are interesting and highlight different aspects of the components of this framework.

Areas of Improvement:
- While the current paper claims that the results easily extend to any dimension, it would make a better paper to include the full result somewhere (even in Appendix). The theorems also assume that the distributions are supported on [0,1]. I am guessing that generalizing beyond [0,1] support will include a diameter term $D$ which may influence the comparison of (4.3) to the eq. in line 227 (probably the last summand).
- (Nit) As a reader, it would have helped if there was an algorithm outline that listed each step of the full framework, even if it appears in the Appendix. For instance, it appears that a time-conditional classifier (between data from $q_0$ and noisy $p_0$) is used to identify the appropriate noise level so that $q_0$ closely approximates a smoothed version of $p_0$ (so to obtain $t_n^{min}$) but another classifier is also used to classify crops of images as coming from $p_0$ or $\tilde{p}_0$ to obtain $t_n^{max}$. While I understood that just “classifier” usually referred to the former and “crops” referred to the latter, it would disambiguate some things (e.g., the details of the classifier in Section F). It would also help me understand the increases in computational cost (e.g., at some point, several crops must be generated I am guessing).
- (Nit) I appreciated the note about the donut paradox, but the text seemed a bit out of place. There was no discussion about mitigation or ablation studies to explore the effect of this donut paradox on the model performance. I wouldn’t say it’s crucially needed but it would give a more complete picture if the authors could comment on it a bit more than the data availability figure.
- I also think more comparisons (even to the previous Ambient Diffusion work) would help the experimental results.

Misc:
- Should capitalize “Gaussian” in line 117
- I would avoid manually shrinking caption text (Figure 1)

 Questions:
- Is Theorem 4.1 new? While the proof is given in the Appendix, the statement seems like something that exists somewhere.
- I have the same question for Theorem 4.2, which looks like a strong data processing inequality.

---

### Official Review · Reviewer_59AN · 2025-09-19

**Rating:** 7
**Confidence:** 2

**Review:**

**Summary:** The paper presents a principled framework for training diffusion models that can effectively utilize low-quality, synthetic, and out-of-distribution images alongside high-quality data. The key insight is that noise addition during the diffusion process naturally dampens the distributional differences between high and low-quality data, making corrupted samples useful for training at appropriate noise levels. The method employs time-conditional classifiers to automatically determine optimal noise assignments for different data quality levels and demonstrates state-of-the-art results on ImageNet generation and improved text-to-image synthesis.

### **Strengths**

* The paper provides rigorous theoretical justification (Theorems 4.1-2) showing how noise addition contracts distributional distances and formalizing the bias-variance tradeoff when using mixed-quality data. This theoretical grounding distinguishes the work from heuristic approaches.

* Rather than simply filtering or ignoring low-quality data, the framework systematically determines when and how to use such data. I like the use of time-conditional classifiers to automatically annotate data with appropriate noise levels, which avoids manual hyperparameter tuning.

* The evaluation spans multiple domains (CIFAR-10, FFHQ, ImageNet, text-to-image) and corruption types (Gaussian blur, JPEG compression, motion blur), with both synthetic and real-world low-quality data. The results consistently show improvements over both naive training and filtering approaches.

### **Areas for Improvement**

**Limited corruption coverage**: While the method works well for high-frequency corruptions (blur, compression), it struggles with low-frequency degradations like masking, color shifts, or fog. This limitation reduces its general applicability, though the authors acknowledge this.

**Computational overhead**: The approach requires training additional classifiers for annotation, increasing the overall computational cost. The authors may mention that this can be avoided with hand-picked annotations.

**The theoretical analysis can be extended.** The theory focuses on 1D cases and Gaussian corruptions. Extensions to higher dimensions and more complex degradation models would naturally strengthen the theoretical foundation.

**"Donut paradox" handling**: The method cannot use some data samples in the intermediate noise range, creating a gap in data availability. While acknowledged, limited analysis is provided on the practical impact of this limitation.

### **Minor Issues**

- Figure 2's illustration of the merging point could be clearer with better labeling
- Some notation could be simplified (e.g., the distinction between various σ parameters)
- The relationship between the two main algorithmic components (high-noise and low-noise regimes) could be better integrated